# Convergence beyond the over-parameterized regime using Rayleigh quotients

**David A. R. Robin**
INRIA
École Normale Supérieure
PSL Research University
david.a.r.robin@gmail.com

**Kevin Scaman**
INRIA
École Normale Supérieure
PSL Research University
kevin.scaman@inria.fr

**Marc Lelarge**
INRIA
École Normale Supérieure
PSL Research University
marc.lelarge@ens.fr

## Abstract

In this paper, we present a new strategy to prove the convergence of deep learning architectures to a zero training (or even testing) loss by gradient flow. Our analysis is centered on the notion of Rayleigh quotients in order to prove Kurdyka-Łojasiewicz inequalities for a broader set of neural network architectures and loss functions. We show that Rayleigh quotients provide a unified view for several convergence analysis techniques in the literature. Our strategy produces a proof of convergence for various examples of parametric learning. In particular, our analysis does not require the number of parameters to tend to infinity, nor the number of samples to be finite, thus extending to test loss minimization and beyond the over-parameterized regime.

## 1 Introduction

In order to understand the performance of vastly over-parameterized networks, various works have investigated the properties of neural tangent kernels (NTK, see Jacot et al., 2018) and their eigenspaces. While the study of these spectra has led to proofs of convergence to global minima despite the non-convexity of the problem, these analyses typically rely on an over-parameterization assumption, or even infinite-width limits, casting a shadow on their applicability. Positive-definiteness of the NTK in particular, granted by the infinite-width limit, does not hold with finite width and a growing number of samples, despite observed successes of neural networks in this regime. We provide a (toy) counter-example in dimension two to better outline this issue, and fix this flaw by re-centering the discussion on Rayleigh quotients, corresponding to fixed directions, rather than positive definiteness, i.e. uniformly bounding in all directions. We give several ideas to obtain bounds on Rayleigh quotients, and provide non-trivial examples for each of the presented ideas, including a recovery of known results, but also a new convergence speed guarantee for the multi-class logistic regression.

**Overview.** In a typical supervised learning task, one is given a training dataset of $n \in \mathbb{N}$ labeled samples $\mathcal{D} = ((x_i, y_i) \in \mathbb{R}^d \times \mathbb{R})_{i \in [n]}$, and a parametric model with $m \in \mathbb{N}$ parameters, $f : \mathbb{R}^m \times \mathbb{R}^d \to \mathbb{R}$. The task is to find parameters fitting the training data, i.e. find $\theta^* \in \mathbb{R}^m$ such that $\forall i \in [n], f(\theta^*; x_i) \approx y_i$. Aggregating these into a single vector $F : \theta \mapsto f_\theta = (f(\theta; x_i))_{i \in [n]}$, this becomes a satisfaction of a system of equations $F(\theta) \approx y \in \mathbb{R}^n$. After choosing a functional loss $\ell : \mathbb{R}^n \to \mathbb{R}_+$, one can learn the associated parameters by gradient flow $\partial_t \theta = -DF(\theta)^T \cdot \nabla \ell(F(\theta))$, where the jacobian of the parameterization $F$ is a matrix $DF(\theta) \in \mathbb{R}^{n \times m}$. This corresponds exactly to the usual practice of defining a parametric function $F$, a functional loss $\ell$, and training by gradient flow on the parameters to minimize the parametric loss $\mathcal{L} = \ell \circ F$. The question is then when does this algorithm converge, and how fast ? Our focus is on the regime of finitely many parameters ($m \in \mathbb{N}$) and large data ($n \to +\infty$), where the over-parameterization arguments ($m \gg n$) are insufficient.

**Context.** Early arguments for the proof of convergence of this system to a loss of zero revolved around strong convexity hypotheses on the loss [see Boyd and Vandenberghe, 2004, Section 9.3.1]. However the parameterization $F$, typically as a neural network, leads to non-convex parametric losses $\mathcal{L}$ even when the functional loss $\ell$ is convex, sometimes even parametric losses that are not locally quasi-convex [for details, see Liu et al., 2022]. Recently, a common solution has been the leverage of Polyak-Łojasiewicz inequalities $\|\nabla\mathcal{L}(\theta)\|_2^2 \geq \mu\,\mathcal{L}(\theta)$, which grant linear convergence by integrating with Grönwall's lemma since for gradient flows it holds $-\partial_t\mathcal{L}(\theta) = \|\nabla\mathcal{L}(\theta)\|_2^2$ (thus for $\mu \in \mathbb{R}_+^*$, $-\partial_t\mathcal{L}(\theta) \geq \mu\,\mathcal{L}(\theta) \Rightarrow \mathcal{L}(\theta_t) \leq \mathcal{L}(\theta_0)\exp(-\mu t)$). For examples in continuous time, see Chizat [2020, Theorem 3.3 and 3.4]. Other results with discrete time include Arora et al. [2019, Theorem 4.1], Oymak and Soltanolkotabi [2019, Theorem 2.1], Liu et al. [2020, Theorem 5.1] and Liu et al. [2022, Eq (3)] . Generally speaking, discretized versions with sufficiently small learning rate have very similar dynamics, at the cost of some local smoothness assumption, and similarly, stochastic versions can leverage the same Łojasiewicz inequalities to prove convergence rates, so the continous-time dynamics proof can be viewed as a first step in the analysis of these more complex cases. These inequalities ensure that there are no critical points that are not global minima, and can hold even for non-convex losses $\mathcal{L}$, although they can be hard to prove.

The behavior of the dynamical system $\partial_t\theta = -\nabla\mathcal{L}(\theta)$ has been shown to be closely tied with the eigenspaces of the Neural Tangent Kernel (NTK) matrix $K(\theta) = DF(\theta) \cdot DF(\theta)^T \in \mathbb{R}^{n \times n}$, introduced in Jacot et al. [2018, Section 4]. More precisely, the local decrease of the loss is $-\partial_t\mathcal{L}(\theta) = \nabla\ell(f_\theta)^T \cdot K(\theta) \cdot \nabla\ell(f_\theta)$. As an example, for the quadratic loss, the gradient satisfies $\|\nabla\ell(f_\theta)\|_2^2 = 4\ell(f_\theta) = 4\mathcal{L}(\theta)$, such that a positive definiteness condition $K(\theta) \succeq \mu > 0$ guarantees the Polyak-Łojasiewicz condition $-\partial_t\mathcal{L}(\theta) \geq 4\mu\,\mathcal{L}(\theta)$, and thus by integration, convergence to zero with a linear convergence speed. Several works, starting with Jacot et al. [2018, Proposition 2] but also Du et al. [2018], have shown that the smallest eigenvalue of this $K(\theta)$ operator is indeed strictly positive if the network is sufficiently overparameterized ($m \gg n$). Subsequent papers have also anayzed *how overparameterized* the network needs to be for this argument to hold, with interesting asymptotic bounds on the number of parameters required [Ji and Telgarsky, 2020, Chen et al., 2021].

**Challenges.** However, this argument for convergence is bound to fail when there are fewer parameters than datapoints ($m < n$). In particular, for a fixed number of parameters $m \in \mathbb{N}$, it is impossible to have both $n \to +\infty$ and $\lambda_{\min}(K(\theta)) > 0$, since $K(\theta) \in \mathbb{R}^{n \times n}$ has rank $m < n$ by definition. As argued by Liu et al. [2022, Proposition 3] for the quadratic loss ($\ell : f \mapsto \|f - y\|_2^2$, satisfying $\nabla\ell(f_\theta) = 2(f_\theta - y) \in \mathbb{R}^n$), this implies that for underparameterized systems, the Łojasiewicz condition cannot be satisfied for all $y$, since $\inf_{u \in \mathbb{R}^n} u^T K(\theta)u/u^T u = \lambda_{\min}(K(\theta)) = 0$. Nonetheless, if some knowledge $y_i = f^*(x_i)$ for some $f^* \in \mathcal{F}_0$ is available, then it is sufficient to show that $\inf_{u \in \mathcal{Y}_0} u^T K(\theta)u/u^T u > 0$, where $\mathcal{Y}_0 = \{(f^*(x_i) - f_\theta(x_i))_i, f^* \in \mathcal{F}_0\} \subseteq \mathbb{R}^n$ is only a subset of the responses $\mathbb{R}^n$ on which the smallest eigenvalue of the NTK might be positive. Bounding the eigenvalues of the NTK away from zero is sufficient, but not necessary, and for cases where the smallest eigenvalue is zero, one can bound the Rayleigh quotient of the gradient and enjoy similar guarantees despite the null eigenvalue(s). Although stated differently in their respective context, previous uses of this restricted eigenvalue argument can be found for instance in Nitanda and Suzuki [2019, Assumption A4: response is NTK-separable], or Arora et al. [2019, Section 6, bounded inverse-NTK response] . We show how the argument used in these particular cases can be extended to a broader setting, and introduce tools to make calculations easier and obtain such guarantees.

Rayleigh quotient bounds enable convergence guarantees in the underparameterized regime ($m < n$) and in particular, for fixed number of parameters $m$, the guarantees hold even when the number of datapoints grows ($n \to +\infty$) and the domain becomes continuous. Letting $n \to +\infty$ requires a slightly different formalism than the vectors and matrices used in this introduction, we will therefore use functional spaces in the following, and the usual notations of differential geometry, with parameters in indices for instance. Contrary to results such as Arora et al. [2019], Du et al. [2019], the formulation using functional spaces, from Jacot et al. [2018], extends to the case where datapoints are arbitrarily close and even identical, allowing guarantees on the expected loss with respect to a continuous distribution and not just the empirical loss measured on finitely many well-separated samples. In particular, these conditions need not rely on properties satisfied only with high-probability by random initialization when $m \to +\infty$, they can be proven even for fixed initialization and $m \in \mathbb{N}$.

Lastly, our analysis ties together in a more general framework the convergence arguments formulated in the functional space [Du et al., 2018, 2019] studying dynamics of the network response, and

similar arguments formulated in the parameter space [Li and Liang, 2018, Zou et al., 2020], by centering the work on the singular values of the network differential $DF(\theta) \in \mathbb{R}^{n \times m}$ rather than the functional-space $DF(\theta) \cdot DF(\theta)^T \in \mathbb{R}^{n \times n}$ or parameter-space $DF(\theta)^T \cdot DF(\theta) \in \mathbb{R}^{m \times m}$ kernels.

**Contributions.** We provide definitions in Sec. 2, then present Kurdyka-Łojasiewicz inequalities, Rayleigh quotients, and their link in Sec. 3. We show in Sec. 4.1 that this recovers previously known linear bounds for the quadratic case. We illustrate a two-dimensional counterexample to the NTK positive-definiteness in Sec. 4.2, and how to overcome it with Rayleigh quotients. In Sec. 4.3 we prove a new bound on logistic regression obtained by the same technique. In Sec. 4.4 and Sec. 4.5, we outline arguments of convergence in more realistic settings and highlight future challenges.

## 2 Definitions for gradient flows and neural tangent kernels

Let $\mathcal{X}$ be a set with no particular structure. We consider the problem of learning a target function $f^* : \mathcal{X} \to \mathbb{R}$, by having access only to samples $(x, f^*(x)) \in \mathcal{X} \times \mathbb{R}$, where $x \sim \mathcal{D}$ are random samples from a probability distribution $\mathcal{D}$ on $\mathcal{X}$. Let $\mathcal{F} = \mathbb{R}^{\mathcal{X}}$ be the vector space of functions from $\mathcal{X}$ to $\mathbb{R}$. The setting presented in the introduction corresponds to $\mathcal{X}$ being finite containing the examples $x_i$ so that functions are represented as vectors $f = (f(x_i))_{i \in [n]}$ and $\mathcal{D}$ is the empirical measure on $\mathcal{X}$.

**Definition 2.1.** *A network map is a function $F : \Theta \to \mathcal{F}$, from $\Theta$ a vector space of finite dimension equipped with an inner product $\langle \cdot, \cdot \rangle_\Theta$, to $\mathcal{F}$ equipped with the topology of pointwise convergence.*

To avoid confusions as much as possible, we will reserve lowercase letters $(f, g, h)$ for functions in $\mathcal{F}$, and the uppercase $F$ for network maps. We will usually put the parameters in index, and inputs between parenthesis, so that for $\theta \in \Theta$, the function $f_\theta : \mathcal{X} \to \mathbb{R}$ sends inputs $x \in \mathcal{X}$ to outputs $f_\theta(x) \in \mathbb{R}$. Readers familiar with differential geometry will note that the assumption that $\Theta$ is a vector space is a simplification, and could be relaxed for instance to a differentiable manifold. However, we are interested in easily readable results closest to applications, and this assumption will avoid cumbersome discussions on the parameter manifold's tangent space, and keep results readable with only some background in linear algebra. In all the examples, it is sufficient for our needs to set $\Theta = \mathbb{R}^m$ with canonical inner product and $\|\cdot\|_\Theta = \|\cdot\|_2$, for some number of parameters $m \in \mathbb{N}$.

**Definition 2.2** ($\mathcal{D}$-seminorm). *Any probability distribution $\mathcal{D}$ on $\mathcal{X}$ induces on $\mathcal{F}$ a bilinear symmetric positive semi-definite form $\langle \cdot, \cdot \rangle_\mathcal{D} : \mathcal{F} \times \mathcal{F} \to \mathbb{R}$, defined for $(g, h) \in \mathcal{F} \times \mathcal{F}$ as*

$$\langle g, h \rangle_\mathcal{D} = \mathbb{E}_{x \sim \mathcal{D}} [g(x)h(x)]$$

*The associated seminorm $\|\cdot\|_\mathcal{D} : \mathcal{F} \to \mathbb{R}_+$ is defined as $\|g\|_\mathcal{D}^2 = \langle g, g \rangle_\mathcal{D} = \mathbb{E}_{x \sim \mathcal{D}} [g(x)^2]$.*

This seminorm does not in general separate points, it is therefore not a norm on $\mathcal{F}$. In particular, if $\mathcal{D}$ does not have full support, then there are non-null functions $g \in \mathcal{F}$ with null seminorm $\|g\|_\mathcal{D} = 0$.

**Definition 2.3** (Gradient flow). *A gradient flow with respect to the differentiable loss $\mathcal{L} : \Theta \to \mathbb{R}_+$ is an absolutely continuous curve $\theta : \mathbb{R}_+ \to \Theta$ satisfying the differential equation $\partial_t \theta = -\nabla \mathcal{L}(\theta)$. Additionally, we say that a gradient flow is trivial if $\mathcal{L}(\theta_0) = 0$, since it implies that for all $t$, $\theta_t = \theta_0$. For $\mathcal{U} \subseteq \Theta$, if $\theta : \mathbb{R}_+ \to \Theta$ is a gradient flow such that $\theta(\mathbb{R}_+) \subseteq \mathcal{U}$ then we write just $\theta : \mathbb{R}_+ \to \mathcal{U}$.*

A common choice for regression with target $f^* \in \mathcal{F}$ is the quadratic loss $\mathcal{L} : \theta \mapsto \|F(\theta) - f^*\|_\mathcal{D}^2$.

If a network map $F : \Theta \to \mathcal{F}$ is differentiable for the pointwise convergence, we will write $\mathrm{d}F_\theta : \Theta \to \mathcal{F}$ for the differential of $F$ at $\theta \in \Theta$, with parameters in index for shortness. Evaluation at $x \in \mathcal{X}$ and derivation with respect to $\theta \in \Theta$ commute, easing computations (see Appendix A.2.2). We write the corresponding gradient $\nabla F_\theta : \mathcal{X} \to \Theta$, defined by $\langle \nabla F_\theta(x), \nu \rangle_\Theta = (\mathrm{d}F_\theta \cdot \nu)(x)$ for all $x \in \mathcal{X}$ and $\nu \in \Theta$.

**Definition 2.4** (Neural Tangent Kernel, NTK form). *A differentiable network map $F : \Theta \to \mathcal{F}$ defines at every point $\theta \in \Theta$ a kernel function $K_\theta : \mathcal{X} \times \mathcal{X} \to \mathbb{R}$ as*

$$K_\theta : (x, x') \mapsto \langle \nabla F_\theta(x), \nabla F_\theta(x') \rangle_\Theta$$

*This function induces a bilinear symmetric positive semi-definite form $K_\theta^\star : \mathcal{F} \times \mathcal{F} \to \mathbb{R}$ as*

$$K_\theta^\star(g, h) = \mathbb{E}_{x \sim \mathcal{D}, x' \sim \mathcal{D}} [g(x)K_\theta(x, x')h(x')]$$

In exponent notation, this bilinear form has signature $K_\theta^\star : \mathbb{R}^{\mathcal{X}} \times \mathbb{R}^{\mathcal{X}} \to \mathbb{R}$, while the kernel $K_\theta \in \mathbb{R}^{\mathcal{X} \times \mathcal{X}}$ is an $n \times n$ matrix when $\mathcal{X}$ is finite with $n \in \mathbb{N}$ elements. Importantly, the (primal) kernel $K_\theta$ is independent of the distribution $\mathcal{D}$, while the (dual) kernel form $K_\theta^\star$ changes with $\mathcal{D}$.

**Definition 2.5** ($\mathcal{D}$-compatibility, functional gradient). *A function $\ell : \mathcal{F} \to \mathbb{R}_+$ is said $\mathcal{D}$-compatible if $\forall (f, g) \in \mathcal{F} \times \mathcal{F}$, it holds that $(f = g)$ $\mathcal{D}$-almost everywhere implies $\ell(f) = \ell(g)$.*

*Moreover, if $\ell$ is $\mathcal{D}$-compatible and differentiable, we say $\nabla \ell : \mathcal{F} \to \mathcal{F}$ is a gradient of $\ell$ if it satisfies*

$$\forall (f, g) \in \mathcal{F} \times \mathcal{F}, \ \langle \nabla \ell_f, g \rangle_\mathcal{D} = \mathrm{d}\ell_f(g)$$

This formalizes the idea that the loss depends *only* on the training samples, and the use of a gradient simplifies the following statements. When it exists, the functional gradient is usually not unique, for it is defined only $\mathcal{D}$-almost everywhere. See Appendix A.2.1 for some examples of conditions under which it is well defined (for instance $\mathcal{D}$ has finite support, or $\ell$ is the expectation of a pointwise loss).

# 3 Rayleigh quotients to obtain Kurdyka-Łojasiewicz inequalities

## 3.1 Context: Kurdyka-Łojasiewicz inequalities for convergence

All convergence proofs presented in this paper rely on inequalities introduced by Kurdyka [1998] of the form of Proposition 3.1. These are used for instance to prove finite length of trajectories in dynamical systems (see e.g. Bolte et al. [2007, Corollary 4.1]), and sufficient to prove convergence to a loss of zero even for non-convex losses. We will therefore direct all later efforts to the construction of such inequalities. This was introduced as an extension to the Polyak-Łojasiewicz inequalities for linear convergence [see e.g. Nguyen, 2017, Section 1.3 for examples], to more general dynamics, and the proof of the following proposition is a simple application of the chain rule to $\varphi \circ \mathcal{L}$ (see A.2.3).

**Proposition 3.1** (Convergence by Kurdyka-Łojasiewicz inequality). *Let $\mathcal{U} \subseteq \Theta$. If $\mathcal{L} : \mathcal{U} \to \mathbb{R}_+$ is such that there exists $\mu \in \mathbb{R}_+^*$ and a strictly increasing differentiable function $\varphi : \mathbb{R}_+^* \to \mathbb{R}$ satisfying*

$$\forall \theta \in \mathcal{U}, \ \mathcal{L}(\theta) \neq 0 \Rightarrow \mathrm{d}\varphi_{\mathcal{L}(\theta)} \ \langle \nabla \mathcal{L}(\theta), \nabla \mathcal{L}(\theta) \rangle_\Theta \geq \mu$$

*Then all non-trivial gradient flows $\theta : \mathbb{R}_+ \to \mathcal{U}$ of $\mathcal{L}$ satisfy $\forall t \in \mathbb{R}_+, \ \mathcal{L}(\theta_t) \leq \varphi^{-1} \left( \varphi(\mathcal{L}(\theta_0)) - \mu t \right)$*

Moreover, if such a flow exists, then $\inf_\theta \mathcal{L}(\theta) = 0$ and $\varphi(u) \to -\infty$ if $u \to 0$ (see Appendix A.2.4).

The central idea, similar to the one used in the following sections, is that a desingularizing function $\varphi : \mathbb{R}_+^* \to \mathbb{R}$ transports the loss evolution $\mathcal{L}(\theta) : I \to \mathbb{R}_+^*$ in $\mathrm{dom}(\varphi) = \mathbb{R}_+^*$ to the space $\mathrm{Im}(\varphi) = \mathbb{R}$ where the evolution is easy to understand, since $(\varphi \circ \mathcal{L})(\theta)$ is bounded by an affine function of time. The desingularizing function provides a way to transfer the understanding of the convergence in the image of $\varphi$ back to the domain of $\varphi$, where the loss evolution is a little more complicated. The condition is also sometimes written $\nabla \mathcal{L} \cdot \nabla \mathcal{L} \geq \psi(\mathcal{L})$, where $\psi : \mathbb{R}_+ \to \mathbb{R}_+$ is $(\psi(u))^{-1} = \mathrm{d}\varphi_u$.

For the case of a linear convergence speed guarantee, the Polyak-Łojasiewicz condition from the introduction (i.e. $-\partial_t \mathcal{L}(\theta) = \|\nabla \mathcal{L}(\theta)\|_2^2 \geq \mu \mathcal{L}(\theta)$) corresponds to the choice $\varphi : u \mapsto \log(u)$. To accurately describe systems with more intricate dynamics, more complicated choices of $\varphi$ may be necessary, see the case of logistic regression in Sec. 4.3 for one such example.

## 3.2 Contribution: Kurdyka-Łojasiewicz inequalities by composition

**Definition 3.2** (Rayleigh quotients of bilinear maps). *Let $(V, \|\cdot\|_V)$ and $(W, \|\cdot\|_W)$ be two vector spaces equipped with seminorms, and let $A : V \times W \to \mathbb{R}$ be a bilinear map. Then for $(x, y) \in V \times W$ such that $(\|x\|_V \in \mathbb{R}_+ \setminus \{0\})$, and $(\|y\|_W \in \mathbb{R}_+ \setminus \{0\})$, define the Rayleigh quotient*

$$\mathrm{R}(A; x, y) = \frac{A(x, y)}{\|x\|_V \|y\|_W}$$

With a symmetric map $A : V \times V \to \mathbb{R}$, the Rayleigh quotient $\mathrm{R}(A; x, x)$ is a convex combination of the eigenvalues of $A$ (which are real-valued), whose weighting depends on $x$. Moreover, the minimal value is attained when $x$ is an eigenvector corresponding to the minimal eigenvalue, and $\lambda_{\min}(A) = \inf_{x \in V \setminus \{0\}} R(A; x, x)$. Lastly, when the map is an inner product, then the Rayleigh quotient $R(\langle \cdot, \cdot \rangle_\Theta; a, b) = \langle a, b \rangle_\Theta / \|a\|_\Theta \|b\|_\Theta$ is a form of cosine similarity. The most common usage is with $x = y$, but the asymmetric definition will be necessary later for the variational bound.

**Proposition 3.3** (Kurdyka-Łojasiewicz inequality by composition)**.** *Let* $F : \Theta \to \mathcal{F}$ *be a differentiable network map, and* $K_\theta$ *the associated neural tangent kernel (by Def 2.4). Let* $\mathcal{U} \subseteq \Theta$ *be a subset of parameters and* $\mathcal{F}_\mathcal{U} = F(\mathcal{U}) \subseteq \mathcal{F}$ *its image by* $F$. *Let* $\ell : \mathcal{F}_\mathcal{U} \to \mathbb{R}_+$ *be a* $\mathcal{D}$-*compatible differentiable loss with gradient* $\nabla\ell : \mathcal{F}_\mathcal{U} \to \mathcal{F}$ *whose seminorm is finite* $\forall f \in \mathcal{F}_\mathcal{U}, \|\nabla\ell_f\|_\mathcal{D} < +\infty$. *Assume that there exists a strictly increasing differentiable* $\varphi : \mathbb{R}_+^* \to \mathbb{R}$ *satisfying*

$$\forall f \in \mathcal{F}_\mathcal{U},\ \ell(f) \neq 0 \Rightarrow \mathrm{d}\varphi_{\ell(f)} \langle \nabla\ell_f, \nabla\ell_f \rangle_\mathcal{D} \geq 1$$

*If the* $K_\theta^\star$-*Rayleigh quotient of the gradient of* $\ell$ *is bounded below, i.e. if there exists* $\mu \in \mathbb{R}_+^*$ *such that*

$$\forall \theta \in \mathcal{U},\ \ell(F(\theta)) \neq 0 \Rightarrow \mathrm{R}\left(K_\theta^\star; \nabla\ell_{F(\theta)}, \nabla\ell_{F(\theta)}\right) \geq \mu$$

*Then, for* $\mathcal{L} = (\ell \circ F) : \mathcal{U} \to \mathbb{R}_+$, *it holds*

$$\forall \theta \in \mathcal{U},\ \mathcal{L}(\theta) \neq 0 \Rightarrow \mathrm{d}\varphi_{\mathcal{L}(\theta)} \langle \nabla\mathcal{L}(\theta), \nabla\mathcal{L}(\theta) \rangle_\Theta \geq \mu$$

The proof of this statement is deferred to Appendix A.3.1, and similar to the usual NTK arguments. If $K_\theta^\star$ is $\mu$-uniformly conditioned, then in particular $K_\theta^\star(\nabla\ell_f, \nabla\ell_f) \geq \mu\langle\nabla\ell_f, \nabla\ell_f\rangle_\mathcal{D}$, which is exactly the Rayleigh quotient condition. The main difference is that it is not necessary to require *uniform* conditioning, it is sufficient for this property to hold on any subspace containing the gradient (and in particular the one-dimensional subspace defined by the gradient, i.e. the Rayleigh quotient).

Kurdyka-Łojasiewicz (KŁ) inequalities provide a reasonable path to convergence bounds, outside the usual convex framework. However, they can still be very difficult to obtain. This proposition splits the parametric-space KŁ inequality into a functional-space KŁ inequality which is easier to obtain (trivial for quadratic losses, see Sec. 4.1; available for cross-entropy for instance, see Sec. 4.3) and a Rayleigh quotient bound, which is the focus of the following propositions. Similarly, we provide hereafter several variational forms that can help break the Rayleigh quotient bounding problem down into smaller blocks that can be easier to compute independently before reassembling.

**Proposition 3.4** (Variational bound)**.** *Let* $F : \Theta \to \mathcal{F}$ *be a differentiable network map,* $K_\theta$ *the associated neural tangent kernel (by Def 2.4), and* $\theta \in \Theta$. *If* $h \in \mathcal{F}$ *satisfies* $\|h\|_\mathcal{D} \neq 0$, *then it holds*

$$\mathrm{R}(K_\theta^\star; h, h) = \sup_{\nu \in \Theta\setminus\{0\}} \mathrm{R}(\mathrm{d}F_\theta^\star; \nu, h)^2$$

*Where* $\mathrm{d}F_\theta^\star$ *is the bilinear form* $(\nu, h) \mapsto \langle \mathrm{d}F_\theta \cdot \nu, h \rangle_\mathcal{D}$ *associated with the linear operator* $\mathrm{d}F_\theta$.

This property is particularly useful to avoid dealing with the square of the differential, and instead obtain lower-bounds on the Rayleigh quotient by carefully selecting (suboptimal) inputs $\nu \in \Theta \setminus \{0\}$.

**Proposition 3.5** (Split cosine - singular value)**.** *Let* $F : \Theta \to \mathcal{F}$ *be a differentiable network map,* $K$ *the associated neural tangent kernel,* $\theta \in \Theta$, *and* $h \in \mathcal{F}$ *such that* $\|h\|_\mathcal{D} \neq 0$. *If there exists a subspace* $\Theta_0 \subseteq \Theta$ *and some* $\mu \in \mathbb{R}_+^*$ *such that there exists* $\nu \in \Theta_0$ *satisfying* $\mathrm{R}(\langle\cdot,\cdot\rangle_\mathcal{D}; \mathrm{d}F_\theta \cdot \nu, h) \geq \mu$, *then for* $\lambda = \inf_{\nu \in \Theta_0} \|\mathrm{d}F_\theta \cdot \nu\|_\mathcal{D}^2 / \|\nu\|_\Theta^2 \in \mathbb{R}_+$, *it holds* $\mathrm{R}(K_\theta^\star; h, h) \geq \mu^2 \lambda$.

This proposition is a trivial consequence of the following one, but is easier to parse while still making apparent the distinction between a geometric quantity $\mu$ and the singular value $\lambda$. See Sec. 4.2 for an example in dimension two, where $\mu$ is defined only by the angle between the gradient and the lemniscate's tangent, independently of the parameterization. Observe on the other hand that as $\lambda$, the speed at which the lemniscate is traveled, changes, so does the gradient flow's convergence speed.

**Proposition 3.6.** *Let* $F : \Theta \to \mathcal{F}$ *be a differentiable network map,* $\theta \in \Theta$, *and* $h \in \mathcal{F}$ *s.t.* $\|h\|_\mathcal{D} \neq 0$.

*Let* $k \in \mathbb{N}^*$. *Let* $(a_i)_{i\in[k]} \in (\Theta \setminus \{0\})^k$ *and* $(g_i)_{i\in[k]} \in (\mathcal{F} \setminus (\|\cdot\|_\mathcal{D})^{-1}(0))^k$. *If* $h \in \mathrm{Span}(g)$, *then*

$$\max_{\nu \in \mathrm{Span}(a)\setminus\{0\}} \mathrm{R}(\mathrm{d}F_\theta^\star; \nu, h) \geq \frac{\lambda_{\min}\left(\mathrm{R}\left(\langle\cdot,\cdot\rangle_\mathcal{D}; \mathrm{d}F_\theta \cdot a_i, g_j\right)_{i,j}\right) \min_{i\in[k]}\|\mathrm{d}F_\theta \cdot a_i\|_\mathcal{D}/\|a_i\|_\Theta}{\sqrt{\lambda_{\max}\left(\mathrm{R}\left(\langle\cdot,\cdot\rangle_\Theta; a_i, a_j\right)_{i,j}\right) \lambda_{\max}\left(\mathrm{R}\left(\langle\cdot,\cdot\rangle_\mathcal{D}; g_i, g_j\right)_{i,j}\right)}}$$

*where the smallest singular value of* $A \in \mathbb{R}^{k\times k}$ *is* $\lambda_{\min}(A) = \min_{u\neq 0} u^T A u / u^T u$ *(resp.* $\max$*).*

If the vectors $(a, g)$ are taken orthogonal and such that $\mathrm{d}F_\theta \cdot a_i = \sigma_i g_i$ for some $\sigma_i \in \mathbb{R}$, then the three matrices are the identity, and only the minimal Rayleigh quotient remains. If they are chosen only approximately orthogonal, then a corresponding multiplicative penalty is incurred.

The proofs of the preceding three propositions are deferred to Appendix A.3.2, A.3.3 and A.3.4 respectively.

# 4 Case studies

## 4.1 Linear models with quadratic loss, recovering known bounds

As a sanity check and simple first contact with the variational bound, we consider a model linear in its parameters, with quadratic loss, and recover the (known optimal) linear convergence rate. This proposition is the continuous time form of Karimi et al. [2016, Theorem 1].

**Proposition 4.1** (Convergence of quadratic-loss linear models). *Let $\mathcal{X} = \Theta = \mathbb{R}^d$, and $F : \Theta \to \mathcal{F}$, be the linear network map $F : \theta \mapsto f_\theta$ defined by $f_\theta(x) = \langle x, \theta \rangle$. Let $f^* : \mathcal{X} \to \mathbb{R}$ be a linear function. Let $\mathcal{L} : \Theta \to \mathbb{R}_+$ be the quadratic loss $\mathcal{L} : \theta \mapsto \|F(\theta) - f^*\|_{\mathcal{D}}^2$ where $\mathcal{D}$ a distribution over $\mathcal{X}$ such that $\mathcal{L}$ is well-defined and finite.*

*If $\theta : \mathbb{R}_+ \to \Theta$ is a gradient flow of $\mathcal{L}$, then for all $t \in \mathbb{R}_+$, it holds $\mathcal{L}(\theta_t) \leq \mathcal{L}(\theta_0) e^{-4 \lambda_{\min}^+(A) t}$, where $A = \mathbb{E}_{x \sim \mathcal{D}} [xx^T] \in \mathbb{R}^{d \times d}$ is the (uncentered) covariance matrix of the samples, and $\lambda_{\min}^+(A)$ its smallest non-null eigenvalue. Moreover, there exists $\mathcal{D}$ such that this bound is an equality.*

The idea is to apply Proposition 3.3. The functional Kurdyka-Łojasiewicz inequality is immediate, and we bound the Rayleigh quotient with Proposition 3.5 applied to the subspace $\Theta_0 = \text{Ker}(A)^\perp$.

*Proof.* Let $\ell : \mathcal{F} \to \mathbb{R}_+, f \mapsto \|f - f^*\|_{\mathcal{D}}^2$ be the functional-space quadratic loss, whose gradient $\nabla \ell_f = 2(f - f^*)$ satisfies the Polyak-Łojasiewicz inequality $\|\nabla \ell_f\|_{\mathcal{D}}^2 \geq 4 \ell(f)$. Hence, let us show $\mathcal{L}(\theta) \neq 0 \Rightarrow \text{R}(K_\theta^\star; \nabla \ell_{F(\theta)}, \nabla \ell_{F(\theta)}) \geq \lambda_{\min}^+(A)$, which is sufficient by applying Proposition 3.3.

Let $\theta^* \in \Theta$ be any parameter such that $f^* = f_{\theta^*}$, where existence is guaranteed by linearity of $f^*$. Observe that the loss can be written $\mathcal{L}(\theta) = (\theta - \theta^*)^T A (\theta - \theta^*)$. Let $\theta \in \Theta$ such that $\mathcal{L}(\theta) \neq 0$. In particular, $\theta - \theta^* \notin \text{Ker}(A)$. Then, let $\Theta_0 = \text{Ker}(A)^\perp$. On one hand, it follows that

$$\sup_{\nu \in \Theta_0 \setminus \{0\}} \frac{\langle \mathrm{d}F_\theta \cdot \nu, F(\theta) - F(\theta^*) \rangle_{\mathcal{D}}^2}{\|\mathrm{d}F_\theta \cdot \nu\|_2^2 \|F(\theta) - F(\theta^*)\|_{\mathcal{D}}^2} = \frac{(u^T A (\theta - \theta^*))^2}{(u^T A u)((\theta - \theta^*)^T A (\theta - \theta^*))} = 1$$

with the maximum attained for $u \in \text{Ker}(A)^\perp \setminus \{0\}$ the orthogonal projection of $(\theta - \theta^*)$ to $\text{Ker}(A)^\perp$, satisfying $A(\theta - \theta^*) = Au$ and $\langle \theta - \theta^*, u \rangle = 0$, thus $u^T A u = u^T A (\theta - \theta^*) = (\theta - \theta^*)^T A (\theta - \theta^*)$.

Then by definition $\inf_{\nu \in \Theta_0 \setminus \{0\}} \|\mathrm{d}F(\theta) \cdot \nu\|_{\mathcal{D}}^2 / \|\nu\|_2^2 = \inf_{\nu \in \Theta_0 \setminus \{0\}} (\nu^T A \nu)/(\nu^T \nu) = \lambda_{\min}^+(A)$. Conclude by Proposition 3.5, with $\mu = 1$ and $\lambda = \lambda_{\min}^+(A)$. Equality is recovered for $A = I_d$. $\square$

This is to be contrasted with a direct proof of the Kurdyka-Łojasiewicz inequality, i.e. showing that

$$\frac{\|\nabla \mathcal{L}(\theta)\|_2^2}{\mathcal{L}(\theta)} = 4 \frac{(\theta - \theta^*)^T A^2 (\theta - \theta^*)}{(\theta - \theta^*)^T A (\theta - \theta^*)} \geq 4 \lambda_{\min}^+(A)$$

Although the proof seems a bit convoluted, the interesting part here is that the original bound can be split into two (hopefully simpler) subproblems, while still allowing the use of knowledge on $(f_\theta - f^*)$, leveraged here by the assumption $(\theta - \theta^*) \in \text{Ker}(A)^\perp$. Note that knowledge of a property such as $(\theta - \theta^*) \in \Theta_0 \subseteq \mathbb{R}^d$ for any subspace $\Theta_0$ could have been used to eliminate any eigenvalues of $A$ on $\Theta_0^\perp$, including strictly positive eigenvalues, there is nothing specific to $\text{Ker}(A)^\perp$ other than the existence of the prior knowledge $(\theta - \theta^*) \notin \text{Ker}(A)$ granted by $\mathcal{L}(\theta) \neq 0$.

## 4.2 Lemniscate-constrained optimization, singular values

We now present a toy example simple enough to allow for explicit computations and constructed to illustrate the importance of parametrization. We consider linear functions in two dimensions where the function $f_{(a,b)} : \mathbb{R}^2 \to \mathbb{R}$, $f_{(a,b)} : (x,y) \mapsto ax + by$ is simply identified with $(a,b) \in \mathbb{R}^2$. We will still consider a quadratic loss but we now assume that the target function $f^* = f_{(a^*, b^*)}$ is linear and with $(a^*, b^*) \in \mathcal{F}_0 = \{(a,b) \in \mathbb{R}^2 \mid (a^2 + b^2)^2 = a^2 - b^2\}$. Although we are looking for a two dimensional linear functions $f^*$, knowing that $f^* \in \mathcal{F}_0$ reduces the "degrees of freedom". In such a scenario in machine learning, we typically incorporate this information in the parametrization. As a result, we now have only one parameter to estimate, i.e. $\Theta = \mathbb{R}$ and our network maps $F : \mathbb{R} \to \mathbb{R}^2$ will satisfy $\overline{\text{Im}(F)} = \mathcal{F}_0$. Note that Bernoulli's lemniscate $\mathcal{F}_0$ (pictured in Fig .2a) is neither a convex

set, nor a manifold (due to the crossing at zero). There is no "natural" parametrization of $\mathcal{F}_0$ and as shown below, the chosen parametrization will matter. For more clarity on the consequences of this parameterization, we use two parameterizations of the lemniscate $\mathcal{F}_0$:

$$F_S : \theta \mapsto \left( \frac{\cos(\theta)}{1 + \sin(\theta)^2}, \frac{\sin(\theta)\cos(\theta)}{1 + \sin(\theta)^2} \right) \quad \text{and,} \quad F_L : \theta \mapsto \left( \frac{1 - \theta^4}{1 + 6\theta^2 + \theta^4}, \frac{2\theta(1 - \theta^2)}{1 + 6\theta^2 + \theta^4} \right).$$

The graph of these parameterizations $\{(\theta, F(\theta)) \mid \theta \in \mathbb{R}\} \subseteq \mathbb{R}^3$ is depicted in Fig. 1. The first, $F_S$ is differentiable $2\pi$-periodic and surjective, satisfying $F_S([0, 2\pi]) = \mathcal{F}_0$. The second, $F_L$ is differentiable, but it is neither injective (since $F_L(-1) = (0,0) = F_L(+1)$) nor surjective. It is a punctured lemniscate $\mathrm{Im}(F_L) = \mathcal{F}_0 \setminus \{(-1,0)\}$, it is only dense in the lemniscate $\overline{\mathrm{Im}(F_L)} = \mathcal{F}_0$.

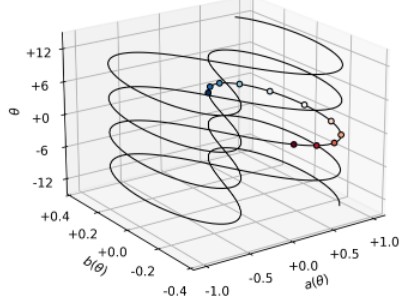
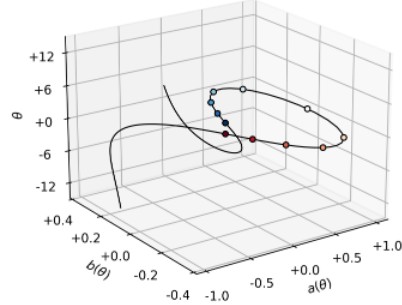

(a) Periodic lemniscate ($F_S$: sphere to lemniscate)    (b) Punctured lemniscate ($F_L$: line to lemniscate)

Figure 1: Graph of the two parameterizations presented (with 11 dots regularly spaced on [-1,+1])

Note that in both cases, the neural tangent kernel $K_\theta$ has rank one (because there is only one parameter), thus $\lambda_{\min}(K_\theta^\star) = 0$ by rank deficiency but we can still prove convergence to zero loss.

To make things even more clear, we assume that all samples are lying on a line: $\mathcal{D}$ is a distribution supported on the one-dimensional subspace $\mathbb{R}\,t$ with $t = (u, v) \in \mathbb{R}^2 \setminus \{0\}$. In words, all the labeled samples are of the form $z(t, a^*u + b^*v) \in \mathbb{R}^2 \times \mathbb{R}$ for some $z \in \mathbb{R}$ and any function $f_{(a,b)}$ with $(a - a^*)u + (b - b^*)v = 0$ will achieve a loss of zero. Indeed as shown in previous section, a standard linear regression in this case converges to a loss of zero but the parameters inferred will not be on the lemniscate $\mathcal{F}_0$. With the parametrization $F_S$ or $F_L$, we will find a solution living on $\mathcal{F}_0$, namely one of the two points in $\ell^{-1}(0) \cap \mathcal{F}_0$, as seen in Figure 2a.

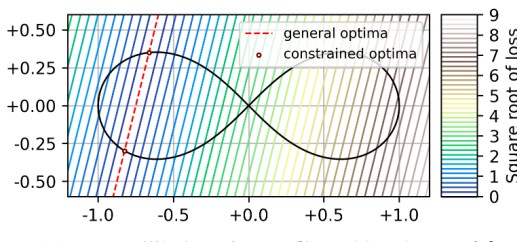
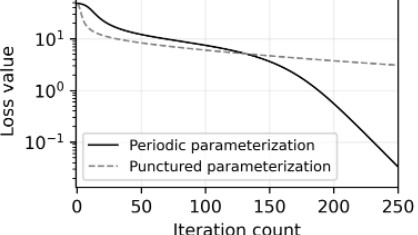

(a) Bernoulli's lemniscate $\mathcal{F}_0$ and level sets of $\ell$

(b) Observed convergence speeds

Figure 2: Loss level sets with parameters $t = (4, -1)$ and $f^*(t) = -3$, corresponding to quadratic loss $\ell : (a, b) \mapsto (4a - b + 3)^2$ and convergence speed with step size $10^{-3}$ and initial estimate $\theta(0) = 0$. Both flows converge to the same functional minimum $(F_S(\theta_S^*) = F_L(\theta_L^*))$, the one depicted on the bottom in (a). Initializing at a different point could have led to a convergence to the other minimum. Proposition 4.2 only shows that the loss converges to zero, leaving unaddressed the question of *which* minimum is reached.

**Proposition 4.2** (Lemniscate convergence with varying speed). *Let $(u, v) \in \mathbb{R}^2$ such that $u > 0$ and $|v| < |u|$. Let $y \in \mathbb{R}_-$ such that the equation $(au + bv = y)$ has exactly two solutions $(a, b) \in \mathcal{F}_0$.*

Let $\ell : \mathbb{R}^2 \to \mathbb{R}_+$ be the quadratic loss $\ell(a,b) \mapsto (au + bv - y)^2$. Let $\theta_S : \mathbb{R}_+ \to \mathbb{R}$ (resp. $\theta_L : \mathbb{R}_+ \to \mathbb{R}$) be a gradient flow with respect to the loss $\ell \circ F_S$ (resp. $\ell \circ F_L$) such that $\theta(0) = 0$. Then there exists a constant $\mu_0 \in \mathbb{R}_+^*$ such that it holds $\ell(F_S(\theta_S(t))) \leq \ell(0) \exp(-4\mu_0^2 \lambda_S^* t)$ and $\ell(F_L(\theta_L(t))) \leq \ell(0) \exp(-4\mu_0^2 \lambda_L^* t)$, where $\lambda_S^* = \frac{1}{2}$ and $\lambda_L^* = \|\nabla F_L(\theta_L^*)\|_2^2$, for $\theta_L^* = \lim_t \theta_L(t)$.

The sketch of this proof is given in Appendix A.4.1. For the numerical values taken in Fig. 2a, we have $\lambda_L^* \approx 4.05 \times 10^{-3}$ showing that our bounds capture the speed of convergence. The idea is as previously, to use the quadratic loss Polyak-Łojasiewicz property ($\|\nabla\ell\|_2^2 \geq 4\ell$) that will grant linear convergence provided we can show $\mathrm{R}(K_\theta^\star; \nabla\ell_{F(\theta)}, \nabla\ell_{F(\theta)}) \geq \mu_0^2 \lambda_S^*$ for all $\theta \in \theta_S(\mathbb{R}_+)$ (resp. $\lambda_L^*$ for $\theta \in \theta_L(\mathbb{R}_+)$), achieved by a variational bound (Proposition 3.4) split according to Proposition 3.5.

## 4.3 Cross-entropy minimization with linear models

We now consider a classification task with $c \geq 1$ classes. Let $\Delta_c = \{u \in (\mathbb{R}_+)^c \mid \sum_{i \in [c]} u_i = 1\}$ be the set of distributions over those classes. The samples $x$ live in $\mathcal{X} = \mathbb{R}^d$ and the target function is $f^* : \mathcal{X} \to \Delta_c$. Let $E : \mathbb{R}^c \to \Delta_c, u \mapsto (\exp(u_i)/\sum_j \exp(u_j))_i$ be the softargmax map. Let $\Theta = \mathbb{R}^{c \times d}$ be the parameter space, and $X : \Theta \to (\mathcal{X} \to \mathbb{R}^c)$ be the operator mapping parameters to linear functions, such that $X(\theta) : x \mapsto \theta \cdot x$. We use the parameterization $F : \theta \mapsto E(X(\theta))$ where $E$ is applied pointwise. For any fixed sample $x \in \mathcal{X}$, we define the loss for this sample as $H_x : \Delta_c \to \mathbb{R}_+, p \mapsto -\sum_{i \in [c]} f_i^*(x) \log(p_i)$. The complete loss used to train this model is then the logistic regression $\mathcal{L} : \theta \mapsto \mathbb{E}_{x \sim \mathcal{D}}[H_x(F(\theta)(x))]$, for which we give a new convergence bound.

$$\Theta \xrightarrow{\ X\ } (\mathcal{X} \to \mathbb{R}^c) \xrightarrow{\ E\ } (\mathcal{X} \to \Delta_c) \xrightarrow{\ H\ } (\mathcal{X} \to \mathbb{R}_+) \xrightarrow{\mathbb{E}_{x \sim \mathcal{D}}} \mathbb{R}_+$$

**Definition 4.3** (Isolation). *A real-valued random variable $Y \in L^1$ is $\kappa$-isolated if $\mathbb{P}(Y \geq \mathbb{E}[Y]) \geq \kappa$.*

All $L^1$ variables are $\kappa$-isolated for some $\kappa > 0$, but we will need a notion of uniform isolation. A random variable $Y$ with finite support, i.e. $\mathbb{P}(Y = y_i) = p_i$ for some $y \in \mathbb{R}^n$ and $p \in \Delta_n$ is $(\min_{i \in [n]} p_i)$-isolated, regardless of the values $y$. This bounds the isolation of the maximal value in a sense. Moreover, if $\psi : \mathbb{R} \to \mathbb{R}$ is increasing and $Y$ is $\kappa$-isolated, then it holds $\mathbb{E}[\psi(Y)] \geq \kappa\psi(\mathbb{E}[Y])$. We use $\kappa = 1/n$ in our experiments (see A.5.6), where $n \in \mathbb{N}^*$ is the number of training points.

**Definition 4.4** (Multi-class separating rays). *We say that a parameter $\zeta \in \mathbb{R}^{c \times d}$ is an $\varepsilon$-separating ray for the distribution $\mathcal{D}$ if it holds for $\mathcal{D}$-almost all $x \in \mathcal{X}$ that*

$$\exists i \in [c], \forall j \in [c] \setminus \{i\}, \ \langle \zeta_i, x \rangle_{\mathbb{R}^d} \geq \langle \zeta_j, x \rangle_{\mathbb{R}^d} + \varepsilon \|\zeta\|_2$$

*where $\zeta_i \in \mathbb{R}^d$ is the $i$-th row of $\zeta$, i.e. if $(\zeta \cdot x) \in \mathbb{R}^c$ has a unique maximum (with a fixed margin).*

This property is invariant by rescaling of $\zeta$ and generalizes the notion of "separation margin" usual in two-class logistic regression. If $\zeta$ is $\varepsilon$-separating for some $\varepsilon > 0$, then for $\mathcal{D}$-almost all inputs $x$, the softargmax classifier $f = F(\zeta) : X \to \Delta_c$ induces a unique label $i \in [c]$ as $i = \arg\max_j f(x)_j$.

**Proposition 4.5** (Convergence speed of logistic regression). *Let $\mathcal{D}$ be a distribution such that the point-loss random variable $\mathcal{L}_x = H_x(f(x))$, where $x \sim \mathcal{D}$, is $\kappa$-isolated for all $f \in F(\Theta)$.*

*Let $\mathcal{L} : \theta \in \Theta \mapsto \mathbb{E}_{x \sim \mathcal{D}}[H_x(F(\theta)(x))] \in \mathbb{R}_+$ be the multi-class cross-entropy loss. If there exists an $\varepsilon$-separating ray $\zeta$ such that $\inf_{\lambda \in \mathbb{R}} \mathcal{L}(\lambda\zeta) = 0$, then for all non-trivial gradient flows $\theta : \mathbb{R}_+ \to \Theta$,*

$$\mathcal{L}(\theta_t) \leq \log\left(1 + \frac{1}{W_0\left(\exp(\varepsilon^2 \kappa^2 t - C)\right)}\right)$$

*where $W_0 : \mathbb{R}_+ \to \mathbb{R}_+$ is the Lambert function, and $C = \log(e^{\mathcal{L}(\theta_0)} - 1) - (e^{\mathcal{L}(\theta_0)} - 1)^{-1} \in \mathbb{R}$.*

The Lambert function $W_0$ is defined by $W_0(x)e^{W_0(x)} = x$, see Corless et al. [1996]. The proof is deferred to Appendix A.5.3. The idea is to prove a functional Kurdyka-Łojasiewicz inequality by leveraging the isolation property, then bound the Rayleigh quotient by leveraging the separation and $\inf \mathcal{L} = 0$ hypotheses to obtain a parametric Kurdyka-Łojasiewicz inequality by Proposition 3.3.

Being a convex problem, the classical argument of Boyd and Vandenberghe [2004] gives a bound $\mathcal{L}(\theta_t) \leq C_0/t$ as long as there is a finite optimum $\theta^* \in \Theta$. This bound becomes vacuous ($C_0 \to +\infty$) in this setting with dirac labels, common in machine learning, because the infimum is located "at

infinity". This assumption has been previously lifted (under separability in Soudry et al. [2018], Nacson et al. [2019] , without separability in Ji and Telgarsky [2019]) to recover the $\mathcal{O}(1/t)$ asymptotic behavior, but without explicit bounds for finite times.

This result is consistent (see Appendix A.5.4) with the asymptotic $\mathcal{O}(1/t)$ bounds from Soudry et al. [2018, Theorem 5] with similar hypotheses, this proposition only makes quantitative the non-asymptotic behavior of this system, and the characteristic quantities driving the convergence speed. To do so, the separation assumption had to be made quantitative, hence the use of $\varepsilon$-separating rays for a fixed positive $\varepsilon$, where previous work used only non-quantified data separation (i.e. $\exists \varepsilon$, $\exists \zeta$ s.t. $\zeta$ is an $\varepsilon$-separating ray for the data), see Appendix A.5.5 for more details. Similarly to the previous section, and contrary to the parameter-direction convergence theorems Soudry et al. [2018, Theorem 5], Nacson et al. [2019, Theorem 3], and Ji and Telgarsky [2019, Theorem 1.1], this proposition does not, on its own, yield any insights on implicit bias (which infimum is reached) towards max-margin rays, additional arguments are required for this purpose. The focus here is on the precise quantification of convergence speed under separability assumptions, with continuous time.

## 4.4 Overparameterized two-layer networks with quadratic loss

Let $\mathfrak{X} = \mathbb{R}^d$, and $\sigma : \mathbb{R} \to \mathbb{R}$ be a non-polynomial Lipschitz map. For $m \in \mathbb{N} \setminus \{0\}$ a number of neurons. Let $\Theta^{(m)} = \mathbb{R}^{m \times d} \times \mathbb{R}^m$ be a parameter set and $F^{(m)} : \Theta^{(m)} \to \mathcal{F}$ be the associated network map $F^{(m)}(w, a) : x \mapsto \sum_{i \in [m]} a_i \, \sigma(w_i \cdot x)$, i.e. a two-layer network[1] with non-linearity $\sigma$.

Let $\mathcal{K} \subseteq \mathfrak{X}$ be compact, and $\mathcal{D}$ a distribution supported on $\mathcal{K}$. Let $f^* \in \mathcal{F}$ be a continuous function. Over $\Theta^{(m)} = \mathbb{R}^{m \times d} \times \mathbb{R}^m$, let $\mathcal{I}_m$ be the (usual in practice) iid normal rescaled initialization with density $p(w, a) = \prod_{i \in [m], j \in [d]} \mathcal{N}(w_{i,j}; 0, 1) \prod_{k \in [m]} \mathcal{N}(a_k; 0, 1/\sqrt{m})$. We write $(x)_+ = \max(0, x)$

**Proposition 4.6.** *Let $\varepsilon \in \mathbb{R}_+^*$, and $\delta \in \,]0, 1[$. There exists $c \in \mathbb{R}_+^*$ such that, for all radii $R \in \mathbb{R}_+^*$, there exists a neuron count $m \in \mathbb{N}$ such that with probability $(1 - \delta)$ over initializations $\theta_0 \sim \mathcal{I}_m$, the quadratic loss $\mathcal{L} : \theta \in \Theta^{(m)} \mapsto \|F^{(m)}(\theta) - f^*\|_{\mathcal{D}}^2$ satisfies the inequality*

$$\forall \theta \in \mathcal{B}(\theta_0, R), \quad \|\nabla \mathcal{L}(\theta)\|_{\Theta}^2 \geq \frac{1}{(\|\theta - \theta_0\|_2 + c)^2} (\mathcal{L}(\theta) - \varepsilon)_+^2$$

*Therefore, for any desired precision $\varepsilon_0 \in \mathbb{R}_+^*$, there exists $(m, \kappa) \in \mathbb{N}^* \times \mathbb{R}_+^*$ such that with probability at least $(1 - \delta)$ over initialization $\theta_0 \sim \mathcal{I}_m$, a gradient flow $\theta : \mathbb{R}_+ \to \Theta$ of $\mathcal{L}$ with $\theta(0) = \theta_0$ satisfies $\forall t \in \mathbb{R}_+$, $\mathcal{L}(\theta_t) \leq \varepsilon_0 + 1/\sqrt[3]{\mathcal{L}(\theta_0)^{-3} + \kappa\, t}$.*

Proof in Appendix A.7. The idea for the proof is to use universal approximation property on compacts [Cybenko, 1989, Barron, 1993, Leshno et al., 1993], to get $\|F(\theta) + \mathrm{d}F_\theta \cdot \nu - f^*\|_{\mathcal{D}}^2 \leq \varepsilon$ for some $\nu \in \Theta$, then derive a Kurdyka-Łojasiewicz inequality from that with a variation of Proposition 3.5. Knowledge of a Kurdyka-Łojasiewicz inequality in a ball around initialization alone is not sufficient to show loss convergence to arbitrary precision in general, but the separable form of this inequality makes it possible, following Scaman et al. [2022, Proposition 4.6]. This proposition shows convergence outside the vastly overparameterized regime ($m$ is finite even with infinite data), but still relies heavily on a (very) large number of neurons. In the next section, we give a partial convergence argument using similar techniques in a much more constrained regime.

## 4.5 Periodic signal recovery

Let $\mathfrak{X} = \mathbb{R}$. Among functions $\mathcal{F} = (\mathbb{R} \to \mathbb{R})$, we are interested in continuous periodic antisymmetric functions, which we parameterize with $\Theta = \mathbb{R}^m \times \mathbb{R}^m$, as $F : \Theta \to \mathcal{F}$, defined for $(a, \omega) \in \Theta$ as $F(a, \omega) : x \mapsto \sum_{i \in [m]} a_i \sin(\omega_i x)$, and $K_{(a, \omega)}$ the associated NTK at the point $(a, \omega) \in \Theta$.

The central property of this application, separating it from the most common machine learning applications, is the inability to obtain good samples. Let $R \in \mathbb{R}_+^*$ be a finite window size, and define the training data distribution $\mathcal{D} = \mathcal{U}(-R, +R)$, the uniform distribution on the interval $[-R, +R]$. Let $\mathcal{F}_0 \subseteq \mathcal{F}$ be the set of continuous periodic antisymmetric functions with period less than $R$. Crucially, we are interested not just in learning the function on the interval, akin to just data retrieval,

---

[1]The bias term usually present in linear layers is omitted to lighten notations, without loss of generality since an additional dimension with non-null constant coordinate can be added to the input domain to compensate for it.

but rather in learning the function in $(\mathbb{R} \to \mathbb{R})$ as a whole. This problem is well defined, i.e. if $f^* \in \mathcal{F}_0$, then $\operatorname{argmin}_{g \in \mathcal{F}_0} \|g - f^*\|_{\mathcal{D}}^2 = \{f^*\}$. The periodicity assumptions makes the data *sufficient* to recover the target function among the hypotheses, however neither the assumption that the training and testing data distributions are identical, nor the assumption that the model has more parameters than there are data points are satisfied. There is infinite data, but there is bias in the sampling.

We will rely on two properties of frequency parameters to show bounds. First, we say that $\omega \in \mathbb{R}^m$ is $\delta$-separated if $\inf_{i \neq j} |\omega_i - \omega_j| \geq \delta$ and $\inf_i |\omega_i| \geq \delta$. Then, we say that the pair $(\omega, \omega^*) \in \mathbb{R}^m \times \mathbb{R}^m$ is $\varepsilon$-paired if $\sup_{i \in [m]} |\omega_i - \omega_i^*| \leq \varepsilon$. Moreover, let $x_0 \in \mathbb{R}_+$ be the first zero of $\operatorname{sinc}''$. ($x_0 \approx 2.0815$).

**Proposition 4.7** (Polyak-Łojasiewicz region)**.** *Let $(\eta, \mu) \in \mathbb{R}_+^* \times \mathbb{R}_+^*$ such that $\eta \leq x_0$ and $\eta < \frac{1}{2}\mu$.*

*Let $f^* \in \mathcal{F}$ be a target, and $\ell : f \in \mathcal{F} \mapsto \frac{1}{2}\|f - f^*\|_{\mathcal{D}}^2$ the quadratic loss, with gradient $\nabla \ell_f = f - f^*$. Assume that there exists $(a^*, \omega^*) \in \Theta$ such that $f^* = F(a^*, \omega^*)$, and $\omega^*$ is $\frac{\mu}{R}$-separated.*

*Then for all $(a, \omega) \in \Theta$ such that $\ell(F(a, \omega)) \neq 0$, $(\omega, \omega^*)$ is $\frac{\eta}{R}$-paired, and $\exists \alpha \in [0, 1], \forall k, a_k^2 \geq \alpha$,*

$$\mathrm{R}\left(K_{(a,\omega)}^\star; \nabla\ell_{F(a,\omega)}, \nabla\ell_{F(a,\omega)}\right) \geq \alpha\left(\phi(\eta) - \frac{1}{\mu - \eta}\right)\frac{(\kappa_0 - \rho_0)^2}{1 + \rho_0}$$

*where with $\psi = -\operatorname{sinc}'$, $\phi = -\operatorname{sinc}''$, and $H = \sum_{k \leq m} \frac{1}{k} \leq 1 + \log(m) \in \mathbb{R}_+$, the constants are*

$$\kappa_0 = \frac{\phi(\eta) - \frac{1}{\mu - \eta}}{\phi(0) + \frac{1}{\mu - \eta}} \qquad \rho_0 = \frac{\psi(\eta) + \frac{1}{\mu - \eta} + \frac{4H}{\mu - 2\eta}}{\phi(\eta) - \frac{1}{\mu - \eta}}$$

*Moreover, $\exists \mu_0 \in \mathbb{R}_+, \forall \mu > \mu_0, \exists \eta > 0, \text{ s.t. } \kappa_0 > \rho_0$. (non degeneracy if enough periods observed)*

Proof in Appendix A.6, leveraging Prop. 3.4 (variational bound) and Prop. 3.6. This shows that when each frequency present in the signal is correctly estimated, then a gradient flow is well-suited for fine-tuning both frequencies and amplitudes. There are sufficiently few interactions to allow each neuron $(a_i, \omega_i)$ to descend towards its target $(a_i^*, \omega_i^*)$. If the modelling hypothesis is verified (the target is a sum of sine waves), there is a finite and small number of neurons giving a sufficiently-parameterized system, and no need to go for vast overparamterization. Letting the number of neurons tend to infinity is one way to ensure there is at least one neuron in each bassin, but not the only way.

## 5 Conclusion

We have shown that Kurdyka-Łojasiewicz inequalities can be leveraged to prove convergence of gradient flows to a loss of zero, even when the convergence speed is not linear. In contrast, Polyak-Łojasiewicz inequalities granted by positive-definiteness of the neural tangent kernel only covered least-squares losses enjoying linear convergence speed. Furthermore, we have shown that by focusing on lowering-bounding Rayleigh quotients rather than all eigenvalues at once, one can prove convergence even when the neural tangent kernel is not positive-definite, the most striking example being the finite-width infinite-data regime, where the neural tangent kernel must have null eigenvalues by rank deficiency. We have provided several simple examples of such convergence proofs outside the vastly over-parameterized regime where there are more parameters than samples, along with tools and preliminary results that lead us to believe that obtaining the crucial Kurdyka-Łojasiewicz inequalities is feasible in more reasonable machine learning settings.

## 6 Acknowledgements

The authors would like to thank Thomas Le Corre, Lucas Weber, and Luca Ganassali for their help with various details of the proofs presented here, along with the anonymous reviewers, for their corrections and help in improving the readability of this work. The authors acknowledge support from the French government under the management of the Agence Nationale de la Recherche as part of the "Investissements d'avenir" program, reference ANR-19-P3IA-0001 (PRAIRIE 3IA Institute).

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
