# A  Appendix

## A.1  Notation summary

Table 1: Notations used in the main text

| | |
|---|---|
| $\mathcal{X}$ | Input of the neural network (viewed as a set with no particular structure) |
| $\mathcal{D}$ | Distribution over $\mathcal{X}$ (may have infinite support) |
| $\mathcal{F} = \mathcal{X}^{\mathbb{R}}$ | Set of $\mathbb{R}$-valued functions on $\mathcal{X}$ |
| $\Theta = \mathbb{R}^d$ | Parameter space of a neural network |
| $\theta \in \Theta$ | Parameters (i.e. weights) of the neural network |
| $\theta_t \in \Theta$ | Parameters at time $t \in \mathbb{R}_+$ when considering a gradient flow $\theta : \mathbb{R} \to \Theta$ |
| $\partial_t \theta_t \in \Theta$ | Time-derivative of the parameters at time $t \in \mathbb{R}_+$ when considering a gradient flow $\theta : \mathbb{R} \to \Theta$ |
| $F : \Theta \to \mathcal{F},\ \theta \mapsto f_\theta$ | Network map, takes weights $\theta$ as input and produces a prediction function $f_\theta : \mathcal{X} \to \mathbb{R}$ as output |
| $\mathrm{d}F_\theta : \Theta \to \mathcal{F},\ \nu \mapsto \mathrm{d}F_\theta \cdot \nu$ | Network map differential at $\theta \in \Theta$, takes weight derivative $\nu$ as input and produces functional derivative $(\mathrm{d}F_\theta \cdot \nu) : \mathcal{X} \to \mathbb{R}$ as output |
| $\langle \cdot, \cdot \rangle_{\mathcal{D}} : \mathcal{F} \times \mathcal{F} \to \mathbb{R}$ | $\langle f, g \rangle_{\mathcal{D}} = \mathbb{E}_{x \sim \mathcal{D}}\left[f(x)g(x)\right]$, see Definition 2.2. |
| $\|\cdot\|_{\mathcal{D}} : \mathcal{F} \to \mathbb{R}_+$ | $\|f\|_{\mathcal{D}} = \sqrt{\mathbb{E}_{x \sim \mathcal{D}}\left[f(x)^2\right]}$, see Definition 2.2. |
| $\ell : \mathcal{F} \to \mathbb{R}_+$ | Functional loss |
| $\mathcal{L} : \Theta \to \mathbb{R}_+$ | Parametric loss, $\mathcal{L} = \ell \circ F$. |
| $K_\theta : \mathcal{X} \times \mathcal{X} \to \mathbb{R}$ $\qquad K_\theta : (x, x') \mapsto \sum_i \partial_{\theta_i} F(\theta)(x)\, \partial_{\theta_i} F(\theta)(x')$ | Neural Tangent Kernel (primal), see Def 2.4 |
| $K_\theta^\star : \mathcal{F} \times \mathcal{F} \to \mathbb{R}$ $\qquad K_\theta^\star : (f, g) \mapsto \mathbb{E}_{x, x' \sim \mathcal{D}}\left[f(x)K_\theta(x, x')g(x')\right]$ | Bilinear form associated with the NTK (dual) |
| $\mathrm{d}F_\theta^\star : \Theta \times \mathcal{F} \to \mathbb{R}$ $\qquad \mathrm{d}F_\theta^\star(\nu, g) = \langle \mathrm{d}F_\theta \cdot \nu, g \rangle_{\mathcal{D}}$ | Bilinear form associated with $\mathrm{d}F_\theta$ and $\mathcal{D}$ |
| $\varphi : \mathbb{R}_+ \to \mathbb{R}$ | Desingularizing function, eases analysis of loss convergence in Proposition 3.1. |
| $\mathrm{d}\varphi : \mathbb{R}_+ \to \mathbb{R},\ u \mapsto \mathrm{d}\varphi_u$ | Derivative of the desingularizing function |
| $R(A; x, y) = \frac{A(x,y)}{\|x\|_V \|y\|_W}$ | Rayleigh quotient at $(x, y) \in V \times W$ of a bilinear map $A : (V, \|\cdot\|_V) \times (W, \|\cdot\|_W) \to \mathbb{R}$ |

## A.2 Details omitted from the main text

### A.2.1 Functional loss gradients

The use of the semi-norm $\|\cdot\|_{\mathcal{D}}$ on the functional space $\mathcal{F}$ comes with some apparent problems, for instance the gradient of the functional loss is not always defined (see Definition 2.5). One solution is to work on a quotient $L^2(\mathcal{D}, \mathbb{R})$ of functions $\mathcal{D}$-almost everywhere identical on which $\|\cdot\|_{\mathcal{D}}$ can be strengthened to a norm. We find this change of space sometimes prone to confusions, for it discards information outside the training region. In the example of the lemniscate from Sec. 4.2, taking the quotient amounts to considering $\mathcal{F}$ to be the line $\mathbb{R}v$ instead of the plane $\mathbb{R}^2$. In particular, the notion of which minimum is reached becomes void because both are identical in the quotient, and the angle between the loss gradient and the lemniscate's tangent is no longer defined.

Instead, we observe that in all reasonable machine learning settings, it seems that the loss has a well-defined gradient with respect to $\langle \cdot, \cdot \rangle_{\mathcal{D}}$ anyway, see e.g. the following proposition

**Lemma A.1.** *Let $\mathcal{U} \subseteq \mathbb{R}$ and $\mathcal{V} \subseteq \mathbb{R}$ be intervals of $\mathbb{R}$. If $\psi : \mathcal{U} \times \mathcal{V} \to \mathbb{R}_+$ is twice continuously differentiable, with derivative with respect to its first variable $\frac{\partial \psi}{\partial u} : \mathcal{U} \times \mathcal{V} \to \mathbb{R}$, and if $\mathcal{D}$ is a distribution over $\mathcal{X} \subseteq \mathbb{R}^d$ with compact support, then for any continuous $f^* : \mathcal{X} \to \mathcal{V}$, the loss*

$$\ell : f \mapsto \mathbb{E}_{x \sim \mathcal{D}} \left[ \psi(f(x), f^*(x)) \right]$$

*is $\mathcal{D}$-compatible, (defined on functions $\mathcal{X} \to \mathcal{U}$ s.t. this expectation is finite), and if $f : \mathcal{X} \to \mathcal{U}$ is continuous, then $\ell$ is differentiable at $f$ and the following is a gradient of $\ell$ at $f$ with respect to $\langle \cdot, \cdot \rangle_{\mathcal{D}}$*

$$\nabla \ell_f : x \mapsto \frac{\partial \psi}{\partial u}(f(x), f^*(x))$$

*Proof.* $\mathcal{D}$-compatibility is immediate. Let $f : \mathcal{X} \to \mathcal{U}$ be continuous, and let $\mathcal{U}_0 \subseteq \mathcal{U}$ be a closed interval such that $f(x) \in \mathcal{U}_0$ holds $\mathcal{D}$-almost surely. Then for all $g : \mathcal{X} \to \mathcal{U}$ such that $f(x) + g(x) \in \mathcal{U}_0$ holds $\mathcal{D}$-almost surely, there exists $R : \mathcal{X} \times [0,1] \to \mathcal{R}$ such that for all $\varepsilon > 0$,

$$
\begin{aligned}
\ell(f + \varepsilon g) &= \mathbb{E}_{x \sim \mathcal{D}} \left[ \psi\left( f(x) + \varepsilon g(x), f^*(x) \right) \right] \\
&= \mathbb{E}_{x \sim \mathcal{D}} \left[ \psi(f(x), f^*(x)) + \frac{\partial \psi}{\partial u} \left( f(x), f^*(x) \right) \varepsilon g(x) + R(x, \varepsilon) \right] \\
&= \ell(f) + \varepsilon \langle \nabla \ell_f, g \rangle_{\mathcal{D}} + \mathbb{E}_{x \sim \mathcal{D}} \left[ R(x, \varepsilon) \right]
\end{aligned}
$$

Moreover, the residual satisfies $R(x, \varepsilon) = o(\varepsilon)$ for all $x$. And for fixed $\varepsilon$, $R(x, \varepsilon)$ is bounded $\mathcal{D}$-almost surely, thus $\mathbb{E}_x \left[ R(x, \varepsilon) \right] = o(\varepsilon)$. Taking the limit when $\varepsilon \to 0$ concludes the proof. $\square$

For instance $\psi : (u, v) \mapsto (u - v)^2$, or $\psi : (u, v) \mapsto -v \log(u)$ if $\mathcal{U} = ]0, 1]$, are relatively common.

In this work, "differentiable" is not taken to imply that the derivative is bounded, for simplicity in the exposition, to avoid dealing with finiteness of involved expectations, since theses issues are entirely orthogonal to our claims, and it is sufficient that a gradient exists for computations to carry out.

### A.2.2 Commutation of evaluation and derivation

For differentiable network map functions $F : \Theta \to \mathcal{F}$, derivation with respect to the parameters in $\Theta$ can be carried out before or after evaluation at $x \in \mathcal{X}$. Formally, if $\bar{\partial}_\theta : (\Theta \to \mathcal{F}) \to (\Theta \to \mathcal{F})$ and $\partial_\theta : (\Theta \to \mathbb{R}) \to (\Theta \to \mathbb{R}))$ are the (functional-valued and real-valued) derivation operators with respect to $\theta \in \Theta$, and if $\delta_x : (\Theta \to \mathcal{F}) \to (\Theta \to \mathbb{R})$ is the evaluation operator at some $x \in \mathcal{X}$ (i.e. $\delta_x(F) : \theta \mapsto F(\theta)(x)$ for all $F : \Theta \to \mathcal{F}$), then $\partial_\theta \circ \delta_x = \delta_x \circ \bar{\partial}_\theta$.

In exponent notation, with $\Theta = \mathbb{R}^d$, the network differential at $\theta \in \Theta$ is a linear function with signature $\mathrm{d}F_\theta : \mathbb{R}^d \to \mathbb{R}^{\mathcal{X}}$. For finite $\mathcal{X}$, it corresponds to a rectangular matrix $\nabla F_\theta \in \mathbb{R}^{\mathcal{X} \times d}$ acting by usual matrix multiplication, with entries $(\partial_{\theta_j} F_\theta(x_i) \in \mathbb{R})$ for $x_i \in \mathcal{X}$ and $j \in [m]$, the partial derivative of the output with respect to the $j$-th parameter, evaluated at the $i$-th point of the dataset.

### A.2.3 Kudyka-Łojasiewicz proof (Proposition 3.1)

*Proof.* Since $\mathcal{L}(\theta_0) \neq 0$, let $I \subseteq \mathbb{R}_+$ be an interval with $0 \in I$, such that $\forall t \in I, \mathcal{L}(\theta_t) > 0$. Over $I$, it holds $\partial_t (\varphi \circ \mathcal{L}(\theta)) = \mathrm{d}(\varphi \circ \mathcal{L})_\theta \cdot \partial_t \theta = \mathrm{d}\varphi_{\mathcal{L}(\theta)} \mathrm{d}\mathcal{L}_\theta \cdot \partial_t \theta = -\mathrm{d}\varphi_{\mathcal{L}(\theta)} \langle \nabla \mathcal{L}(\theta), \nabla \mathcal{L}(\theta) \rangle \leq -\mu$. Thus by integration, $\forall t \in I, \varphi(\mathcal{L}(\theta_t)) - \varphi(\mathcal{L}(\theta_0)) \leq -\mu t$. The result over $I$ follows by inverting $\varphi$, and is extended to times $t \in \mathbb{R}_+$ such that $\mathcal{L}(\theta_t) = 0$ by noticing that $\forall v \in \mathrm{Im}(\varphi), 0 < \varphi^{-1}(v)$. $\qquad \square$

### A.2.4 Kurdyka-Łojasiewicz details

Two assumptions are somewhat hidden in Proposition 3.1. If $\mathcal{L} : \mathcal{U} \to \mathbb{R}_+$ satisfies the Kurdyka-Łojasiewicz inequality $\mathrm{d}\varphi_\mathcal{L} \langle \nabla \mathcal{L}, \mathcal{L} \rangle_\Theta \geq \mu$, and if there exists a gradient flow $\theta : \mathbb{R}_+ \to \mathcal{U}$, then $\inf \mathcal{L} = 0$ and $\varphi(u) \xrightarrow[u \to 0]{} -\infty$.

Let $J = \mathrm{Im}(\varphi) \subseteq \mathbb{R}$. $J$ is an interval, by continuity of $\varphi$. By Proposition 3.1, for all $t \in \mathbb{R}_+$, it holds $\varphi(\mathcal{L}(\theta_t)) \leq \varphi(\mathcal{L}(\theta_0)) - \mu t$. Therefore $\inf(J) \leq \varphi(\mathcal{L}(\theta_0)) - \mu t \to -\infty$ when $t \to +\infty$, hence $\inf(J) = -\infty$. Since $\varphi : \mathbb{R}_+^* \to J$ is strictly increasing, this implies that $\varphi(u) \xrightarrow[u \to 0]{} -\infty$.

By the same proposition, it follows that $\mathcal{L}(\theta_t) \leq \varphi^{-1}(\varphi(\mathcal{L}(\theta_0)) - \mu t)$. But since it holds that $\varphi(\mathcal{L}(\theta_0)) - \mu t \to -\infty$ when $t \to +\infty$, we conclude that $\mathcal{L}(\theta_t) \to 0$, in particular $(\inf \mathcal{L}) = 0$.

While these may be viewed as restrictions of the applicability of Proposition 3.1, we claim that the proof and general ideas are simple enough to be straightforwardly extended to any related setting, the most important part is that the statement is sufficiently clear to convey the idea for the proof.

## A.3 Proofs omitted from the main text

### A.3.1 Proof of composition property (Proposition 3.3)

*Proof of Proposition 3.3.* Let $\theta \in \Theta$, and $f_\theta = F(\theta) \in \mathcal{F}$, such that $\mathcal{L}(\theta) \neq 0$. Let us show that
$$\langle \nabla \mathcal{L}(\theta), \nabla \mathcal{L}(\theta) \rangle_\Theta = \langle \nabla \ell_{f_\theta}, \nabla \ell_{f_\theta} \rangle_\mathcal{D} \, \mathrm{R}\left(K_\theta^\star; \nabla \ell_{f_\theta}, \nabla \ell_{f_\theta}\right)$$

First, the right-hand side is well-defined because $\mathcal{L}(\theta) \neq 0$ implies $\|\nabla \ell_{F(\theta)}\|_\mathcal{D}^2 \neq 0$. Indeed, $\ell(f_\theta) = \mathcal{L}(\theta) \neq 0$, therefore $\mathrm{d}\varphi_{\ell(f_\theta)} \|\nabla \ell_{f_\theta}\|_\mathcal{D}^2 \geq 1$, however $\varphi$ is strictly increasing, so $\mathrm{d}\varphi_{\ell(f_\theta)} > 0$.

Since $\mathcal{L} = \ell \circ F$, it follows that $\nabla \mathcal{L}(\theta) = \mathbb{E}_{x \sim \mathcal{D}}\left[\nabla F_\theta(x) \nabla \ell_{F(\theta)}(x)\right]$. Therefore

$$\langle \nabla \mathcal{L}(\theta), \nabla \mathcal{L}(\theta) \rangle_\Theta = \left\langle \mathbb{E}_{x \sim \mathcal{D}}\left[\nabla F_\theta(x) \nabla \ell_{F(\theta)}(x)\right], \mathbb{E}_{x' \sim \mathcal{D}}\left[\nabla F_\theta(x') \nabla \ell_{F(\theta)}(x')\right]\right\rangle_\Theta \quad (1)$$
$$= \mathbb{E}_{x \sim \mathcal{D}, x' \sim \mathcal{D}}\left[\nabla \ell_{F(\theta)}(x) \langle \nabla F_\theta(x), \nabla F_\theta(x') \rangle_\Theta \nabla \ell_{F(\theta)}\right] \quad (2)$$
$$= \mathbb{E}_{x \sim \mathcal{D}, x' \sim \mathcal{D}}\left[\nabla \ell_{F(\theta)}(x) K_\theta(x, x') \nabla \ell_{F(\theta)}\right] \quad (3)$$
$$= K_\theta^\star\left(\nabla \ell_{F(\theta)}, \nabla \ell_{F(\theta)}\right) \quad (4)$$
$$= \frac{K_\theta^\star\left(\nabla \ell_{F(\theta)}, \nabla \ell_{F(\theta)}\right)}{\langle \nabla \ell_{F(\theta)}, \nabla \ell_{F(\theta)} \rangle_\mathcal{D}} \langle \nabla \ell_{F(\theta)}, \nabla \ell_{F(\theta)} \rangle_\mathcal{D} \quad (5)$$
$$= R\left(K_\theta^\star; \nabla \ell_{F(\theta)}, \nabla \ell_{F(\theta)}\right) \langle \nabla \ell_{F(\theta)}, \nabla \ell_{F(\theta)} \rangle_\mathcal{D} \quad (6)$$

Where $(1)$ is by definition of $\nabla \ell$ (Def 2.5) as cited above, $(2)$ by linearity, $(3)$ by definition of $K(\theta)$ (Def 2.4a), $(4)$ by definition of $K_\theta^\star$ (Def 2.4b), $(5)$ is well defined because $\mathcal{L}(\theta) \neq 0$, and $(6)$ is by definition of R (Def 3.2). The result follows immediately by multiplying both sides by $\mathrm{d}\varphi_{\mathcal{L}(\theta)}$. $\qquad \square$

### A.3.2 Proof of variational bound (Proposition 3.4)

*Proof of Proposition 3.4.* By the variational form of the $\ell_2$-norm induced by the inner product on $\Theta$.

$$K_\theta^\star(h, h) = \mathbb{E}_{x \sim \mathcal{D}, x' \sim \mathcal{D}}\left[h(x)(\nabla F_\theta(x) \cdot \nabla F_\theta(x'))h(x')\right]$$
$$= \|\mathbb{E}_{x \sim \mathcal{D}}\left[\nabla F_\theta(x)h(x)\right]\|_\Theta^2$$
$$= \sup_{\nu \in \Theta \setminus \{0\}} \frac{1}{\|\nu\|_\Theta^2} \langle \nu, \mathbb{E}_{x \sim \mathcal{D}}\left[\nabla F_\theta(x)h(x)\right] \rangle_\Theta^2$$
$$= \sup_{\nu \in \Theta \setminus \{0\}} \frac{1}{\|\nu\|_\Theta^2} \langle \mathrm{d}F_\theta \cdot \nu, h \rangle_\mathcal{D}^2$$

It then suffices to divide both sides by $\langle h, h \rangle_{\mathcal{D}} = \|h\|_{\mathcal{D}}^2 \neq 0$. $\qquad \square$

### A.3.3 Proof of cosine-singular split (Proposition 3.5)

*Proof of Proposition 3.5.* If $\lambda = \inf_{\nu \in \Theta_0 \setminus \{0\}} \| \, \mathrm{d}F_\theta \cdot \nu\|_{\mathcal{D}}^2 / \|\nu\|_\Theta^2 = 0$, then the result is immediate because $\mathrm{R}(K_\theta^\star; h, h) \geq 0$ by positive semi-definiteness. Thus, assume $\lambda > 0$.

$$\mathrm{R}\left(K_\theta^\star; h, h\right) = \sup_{\nu \in \Theta \setminus \{0\}} \mathrm{R}(\, \mathrm{d}F_\theta^\star; \nu, h)^2 \tag{1}$$

$$= \sup_{\nu \in \Theta \setminus \{0\}} \frac{\langle \, \mathrm{d}F_\theta \cdot \nu, h \rangle_{\mathcal{D}}^2}{\|\nu\|_\Theta^2 \|h\|_{\mathcal{D}}^2} \tag{2}$$

$$\geq \sup_{\nu \in \Theta_0 \setminus \{0\}} \frac{\langle \, \mathrm{d}F_\theta \cdot \nu, h \rangle_{\mathcal{D}}^2}{\|\nu\|_\Theta^2 \|h\|_{\mathcal{D}}^2} \tag{3}$$

$$= \sup_{\nu \in \Theta_0 \setminus \{0\}} \frac{\langle \, \mathrm{d}F_\theta \cdot \nu, h \rangle_{\mathcal{D}}^2}{\| \, \mathrm{d}F_\theta \cdot \nu\|_{\mathcal{D}}^2 \|h\|_{\mathcal{D}}^2} \frac{\| \, \mathrm{d}F_\theta \cdot \nu\|_{\mathcal{D}}^2}{\|\nu\|_\Theta^2} \tag{4}$$

$$\geq \left( \sup_{\nu \in \Theta_0 \setminus \{0\}} \frac{\langle \, \mathrm{d}F_\theta \cdot \nu, h \rangle_{\mathcal{D}}^2}{\| \, \mathrm{d}F_\theta \cdot \nu\|_{\mathcal{D}}^2 \|h\|_{\mathcal{D}}^2} \right) \left( \inf_{\nu \in \Theta_0 \setminus \{0\}} \frac{\| \, \mathrm{d}F_\theta \cdot \nu\|_{\mathcal{D}}^2}{\|\nu\|_\Theta^2} \right) \geq \mu^2 \lambda \tag{5}$$

where (1) is Prop 3.4, (2) the definition of R, (3) because the supremum is increasing with respect to inclusion, (4) is well-defined because $\lambda > 0$, and (5) is a uniform bound on the second factor. $\qquad \square$

### A.3.4 Proof of approximate SVD (Proposition 3.6)

*Proof of Proposition 3.6.* Since $h \in \mathrm{Span}(g)$, let $u \in \mathbb{R}^k$ such that $h = \sum_i \frac{u_i}{\|g_i\|_{\mathcal{D}}} g_i$. Then let $\rho = \min_i \| \, \mathrm{d}F_\theta \cdot a_i\|_{\mathcal{D}} / \|a_i\|_\Theta$. If $\rho = 0$ then the proposition is verified: let $\nu \in \mathrm{Span}(a) \setminus \{0\}$, observe either $\mathrm{R}(\, \mathrm{d}F_\theta^\star; \nu, h) \geq 0$, which satisfies the property, or $\mathrm{R}(\, \mathrm{d}F_\theta^\star; -\nu, h) = -\mathrm{R}(\, \mathrm{d}F_\theta^\star; \nu, h) > 0$. Therefore assume in the following that $\rho > 0$. Let $v \in \mathbb{R}^k$ be $v_i = u_i \|a_i\|_\Theta / \| \, \mathrm{d}F_\theta \cdot a_i\|_{\mathcal{D}}$.

$$\max_{\nu \in \mathrm{Span}(a) \setminus \{0\}} \mathrm{R}(\, \mathrm{d}F_\theta^\star; \nu, h)$$

$$\geq \mathrm{R}\left( \, \mathrm{d}F_\theta^\star; \sum_i \frac{u_i}{\| \, \mathrm{d}F_\theta a_i\|_{\mathcal{D}}} a_i, h \right) \tag{1}$$

$$= \frac{\sum_{i,j} \frac{u_i}{\| \, \mathrm{d}F_\theta \cdot a_i\|_{\mathcal{D}}} \frac{u_j}{\|g_j\|_{\mathcal{D}}} \langle \, \mathrm{d}F_\theta \cdot a_i, g_j \rangle_{\mathcal{D}}}{\sqrt{\sum_{i,j} \frac{u_i}{\|a_i\|_\Theta} \frac{\|a_i\|_\Theta}{\| \, \mathrm{d}F_\theta \cdot a_i\|_{\mathcal{D}}} \frac{u_j}{\|a_j\|_\Theta} \frac{\|a_j\|_\Theta}{\| \, \mathrm{d}F_\theta \cdot a_j\|_{\mathcal{D}}} \langle a_i, a_j \rangle_\Theta} \sqrt{\sum_{i,j} \frac{u_i}{\|g_i\|_{\mathcal{D}}} \frac{u_j}{\|g_j\|_{\mathcal{D}}} \langle g_i, g_j \rangle_{\mathcal{D}}}} \tag{2}$$

$$= \frac{\sum_{i,j} u_i u_j \mathrm{R}(\langle \cdot, \cdot \rangle_{\mathcal{D}}; \, \mathrm{d}F_\theta \cdot a_i, g_j)}{\sqrt{\sum_{i,j} v_i v_j \mathrm{R}(\langle \cdot, \cdot \rangle_\Theta; a_i, a_j)} \sqrt{\sum_{i,j} u_i u_j \mathrm{R}(\langle \cdot, \cdot \rangle_{\mathcal{D}}; g_i, g_j)}} \tag{3}$$

$$\geq \frac{\lambda_{\min}(\mathrm{R}(\langle \cdot, \cdot \rangle_{\mathcal{D}}; \, \mathrm{d}F_\theta \cdot a_i, g_j)) \|u\|_2^2}{\sqrt{\lambda_{\max}(\mathrm{R}(\langle \cdot, \cdot \rangle_\Theta; a_i, a_j)) \|v\|_2^2} \sqrt{\lambda_{\max}(\mathrm{R}(\langle \cdot, \cdot \rangle_{\mathcal{D}}; g_i, g_j)) \|u\|_2^2}} \tag{4}$$

$$\geq \frac{\lambda_{\min}(\mathrm{R}(\langle \cdot, \cdot \rangle_{\mathcal{D}}; \, \mathrm{d}F_\theta \cdot a_i, g_j)) \, \min_{i \in [k]} \| \, \mathrm{d}F_\theta \cdot a_i\|_{\mathcal{D}} / \|a_i\|_\Theta}{\sqrt{\lambda_{\max}(\mathrm{R}(\langle \cdot, \cdot \rangle_\Theta; a_i, a_j))} \sqrt{\lambda_{\max}(\mathrm{R}(\langle \cdot, \cdot \rangle_{\mathcal{D}}; g_i, g_j))}} \tag{5}$$

where (1) is evaluation of the variational form, (2) by definition of R and bilinearity, (3) is a reorganization by bilinearity, (4) by definition of $(\lambda_{\min}, \lambda_{\max})$, and (5) by using $\|v\|_2 \leq \frac{1}{\rho} \|u\|_2$. $\qquad \square$

## A.4 Computations for the lemniscate

### A.4.1 Convergence on the lemniscate

**Proof sketch for Proposition 4.2.** As previously, the quadratic loss satisfies a Polyak-Łojasiewicz property ($\|\nabla \ell\|_2^2 \geq 4\ell$) that will grant linear convergence provided we can show a lower bound for $\mathrm{R}(K_\theta^\star; \nabla \ell_{F(\theta)}, \nabla \ell_{F(\theta)})$ for all $\theta(t)$ for each parameterization, which we will achieve by a variational bound (Proposition 3.4), then splitting the variational term according to Proposition 3.5.

We start by computing in closed form the differentials of each parameterization.

$$\nabla F_S(\theta) = \left( -\sin(\theta)\frac{((1+\sin^2(\theta)) + 2\cos^2(\theta))}{(1+\sin^2(\theta))^2}, \frac{-\sin^4(\theta) - \sin^2(\theta) + (1-\sin^2(\theta))\cos^2(\theta)}{(1+\sin^2(\theta))^2} \right)$$

$$\nabla F_L(\theta) = \frac{1}{(\theta^4 + 6\theta^2 + 1)^2} \left( -4\theta(3\theta^4 + 2\theta^2 + 3), 2(\theta^6 - 9\theta^4 - 9\theta^2 + 1) \right)$$

Without loss of generality, assume $v \geq 0$ (by symmetry). Now for both parameterizations, we need to study several functions from $\mathbb{R}$ to $\mathbb{R}$. By Proposition 3.4 then Proposition 3.5,

$$R(K_{S,\theta}^\star; \nabla\ell_{F_S(\theta)}, \nabla\ell_{F_S(\theta)}) = \sup_{\nu \in \mathbb{R}\setminus\{0\}} R(dF_S(\theta)^\star; \nu, \nabla\ell_{F_S(\theta)})^2$$

$$= \sup_{\nu \in \mathbb{R}\setminus\{0\}} R(\langle\cdot,\cdot\rangle_{\mathbb{R}^2}; dF_s(\theta)\cdot\nu, \nabla\ell_{F_S(\theta)})^2 \times \frac{\|dF_S(\theta)\cdot\nu\|_2^2}{\|\nu\|_2^2}$$

$$= R(\langle\cdot,\cdot\rangle_{\mathbb{R}^2}; \nabla F_s(\theta), \nabla\ell_{F_S(\theta)})^2 \times \|\nabla F_S(\theta)\|_2^2$$

Observe that $\nabla\ell(a,b) = 2(au + bv - y)(u,v) \in \mathbb{R}^2$. By hypothesis, if $(a,b) = \theta(t)$ then $(au + bv - y) \geq 0$ (for both $\theta = \theta_S$ and $\theta = \theta_L$) because this quantity is positive at initialization and cannot change signs (if it becomes null, the loss is null and the flow stops).

Let $\theta_S^* = \min\{\theta \mid \ell(F_S(\theta)) = 0\}$ and $\theta_L^* = \min\{\theta \mid \ell(F_L(\theta)) = 0\}$ be the first zeros of each loss on $\mathbb{R}_+$. Let $\mu_S : \theta \mapsto R(\langle\cdot,\cdot\rangle_{\mathbb{R}^2}; \nabla F_S(\theta), -\nabla\ell_{F(\theta)})$ (respectively $\mu_L$). Show by computation that there exists $\mu_0 \in \mathbb{R}_+^*$ such that $\mu_S(\theta) \geq \mu_0$ for all $\theta \in [0, \theta_S^*[$. This constant is not dependent of the parameterization because if $F_S(\theta) = F_L(\nu)$ then $dF_S(\theta) \in \mathbb{R}_+^* \cdot dF_L(\nu)$. Thus $\mu_L(\theta) \geq \mu_0$ for all $\theta \in [0, \theta_L^*[$. Moreover, $\partial_t\theta_S(t)$ and $\mu_S(\theta_S(t))$ have same sign, and $\theta_S(0) = 0$, so $\theta_S$ is increasing over time (respectively $\partial_t\theta_L \geq 0$ for the other parameterization). See Fig. 3 for an illustration.

Now let $\lambda_S : \theta \mapsto \|dF_S(\theta)\|_2^2$ be the corresponding singular value for $F_S$ (respectively $\lambda_L : \theta \mapsto \|dF_L(\theta)\|_2^2$ for $F_L$). Observe that $\lambda_S$ is bounded below on $[0, \theta_s^*[$ (respectively $\lambda_L$ on $[0, \theta_L^*[$). Conclude by lower-bounding the (positive) product with the product of the (positive) lower-bounds.

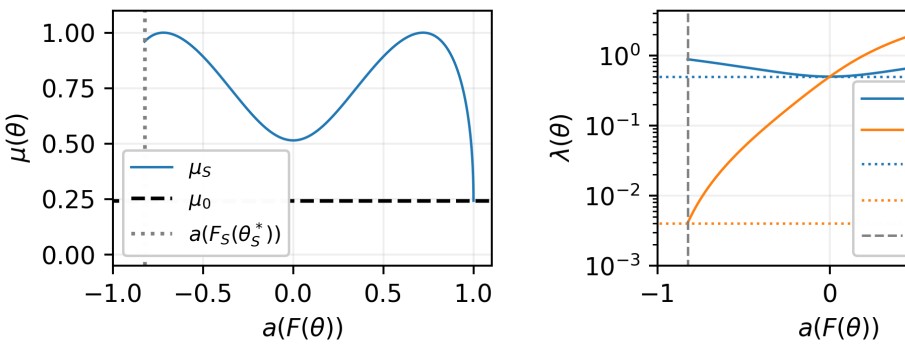

(a) Angle between tangent and gradient     (b) Singular value of lemniscate parameterization

Figure 3: Decomposition of the bound for $(u,v,y) = (4,-1,-3)$, as for Fig. 2a, for $\theta \geq 0$

The additional properties that $\lambda_S^* \geq \frac{1}{2}$ and $\lambda_L^* = \|\nabla F_L(\theta_L^*)\|_2^2$ are depicted on Fig. 3b.

### A.4.2 Convergence speed details predictable from the Rayleigh quotient

We depict in Fig. 4 the gradient flow for the "sphere to lemniscate" ($F_S$) parameterization (already depicted in Fig. 2b), and show that the slowdowns observed in the decrease of the loss correspond to the points at which the gradient of the loss is less aligned with the lemniscate's tangent (corresponding to low values of $\mu_S$). This is because the Rayleigh quotient is $R(K_\theta^\star; \nabla\ell_{F_S(\theta)}, \ell_{F_S(\theta)}) = \mu_S(\theta)^2\lambda_S(\theta)$, and the singular-value factor $\lambda_S$ is almost constant, as can be seen on Fig. 3b.

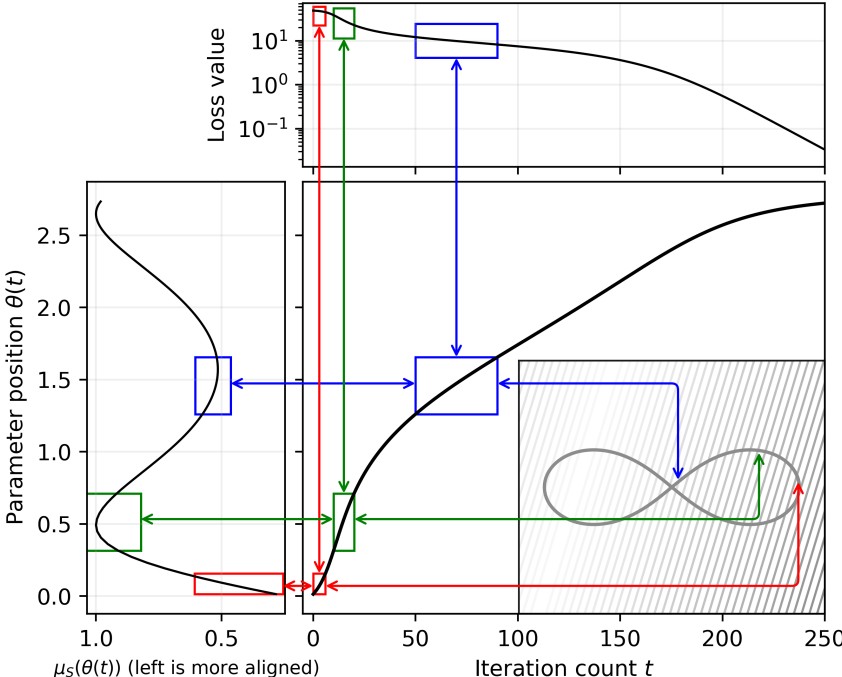

Figure 4: Alignement of gradient and lemniscate's tangent, with consequences on convergence speed (best viewed in color). Red (bottom-most) and blue (top-most) regions correspond to low $\mu_S$ and slowdowns in the loss decrease, Green (middle) region corresponds to higher $\mu_S$ and an acceleration.

### A.5  Computations for logistic regression

#### A.5.1  Pointwise gradients

We use the notations of section 4.3. Let $x \in \mathfrak{X}$ and the corresponding label $y \in \Delta_c$. Recall $\ell_x = H_x \circ E$, where $H_x : p \in \Delta_c \mapsto -\sum_i y_i \log(p_i)$. Let $u \in \mathbb{R}^c$. Let us show that $\nabla \ell_x(u) = E(u) - y$.

*Proof.* The derivative of $H_x$ is straightforward

$$\frac{\partial H_x}{\partial p_i}(p) = -\frac{y_i}{p_i}$$

The derivative of the $i$-th coordinate of softargmax $E_i : u \mapsto \exp(u_i)/\sum_j \exp(u_j)$ is

$$\frac{\partial E_i}{\partial u_j}(u) = \frac{\delta_{i=j} \exp(u_i) \sum_k \exp(u_k) - \exp(u_i) \exp(u_j)}{\left(\sum_k \exp(u_k)\right)^2} = \delta_{i=j} E_i(u) - E_i(u) E_j(u)$$

The result follows by chain rule, using $\sum_{i \in [c]} y_i = 1$

$$\frac{\partial \ell_x}{\partial u_i}(u) = \sum_j \frac{\partial H_x}{\partial p_j}(E(u)) \frac{\partial E_j}{\partial u_i}(u) = \sum_j -\frac{y_j}{E_j(u)} \left(\delta_{i=j} E_j(u) - E_i(u) E_j(u)\right) = E_i(u) - y_i$$

$\square$

#### A.5.2  Separating ray with zero loss implies dirac labels

As a first step, consider the following lemma. Let $x \in \mathfrak{X}$ and $H_x : p \in \Delta_c \mapsto -\sum_i y_i \log(p_i)$. If $p : \mathbb{N} \to \Delta_c$ is a sequence converging to $q \in \Delta_c$ such that $H_x(p(k)) \to 0$, then $q = y$ and

$\exists i, \forall j, y_j = \delta_{i=j}$. To prove this by contradiction, assume there exists $i \neq j$ such that $y_i \neq 0$ and $y_j \neq 0$. Since $H_x(q) < +\infty$, it holds $q_i \neq 0$ and $q_j \neq 0$, thus $\max(q_i, q_j) < 1$ and $H_x(q) \geq -y_i \log(q_i) - y_j \log(q_j) \geq -(y_i + y_j) \log(\max(q_i, q_j)) > 0$ which contradicts $H_x(p(k)) \to 0$, thus $y$ is a dirac. Finally, if $i \in [c]$ is such that $y_i = 1$, then $H_x(p_x(k)) \to 0$ implies $q_i = 1$.

It remains to show that this holds for $\mathcal{D}$-almost all responses $y$. Let $\zeta \in \mathbb{R}^{c \times d}$ be an $\varepsilon$-margin separating ray satisfying $\inf_\lambda \mathcal{L}(\lambda \zeta) = 0$. Let $\lambda : \mathbb{N} \to \mathbb{R}$ be a sequence such that $\mathcal{L}(\lambda_k \zeta) \underset{k \to +\infty}{\longrightarrow} 0$.

For $x \in \mathcal{X}$, the sequence $k \mapsto E(\lambda_k \zeta \cdot x)$ has values in $\Delta_c$, which is compact. Hence extract from it a convergent sequence $(p_x(k))_{k \in \mathbb{N}}$. Then, $(H_x(p_x(k)))_k$ is a sequence of positive random variables converging in expectation to zero, therefore up to extraction of another subsequence, it converges almost surely to zero [see e.g. Gut, 2013, Theorem 3.4, page 212]. Thus, it holds almost surely that $y$ is a dirac and $p_x(k) \to y$. Moreover, for $\mathcal{D}$-almost all $x \in \mathcal{X}$, there exists $i^* \in [c]$ such that for all $j \in [c]$, $\langle \zeta_{i^*}, x \rangle \geq \langle \zeta_j, x \rangle$, hence $p_{x,i^*}(k) \geq p_{x,j}(k)$, which implies $y = (\delta_{j=i^*})_{j \in [c]}$.

### A.5.3 Proof of convergence speed for logisitic regression

For $\mathcal{D}$-almost all $x \in \mathcal{X}$, let $\ell_x = H_x \circ E : \mathbb{R}^c \to \mathbb{R}_+$. For $u \in \mathbb{R}^c$, by a simple calculation, this has gradient $\nabla \ell_x(u) = E(u) - y$ (see appendix A.5.1). Then, define $\ell : (\mathcal{X} \to \mathbb{R}^c) \to \mathbb{R}_+$, as $\ell(u) = \mathbb{E}_{x \sim \mathcal{D}}[\ell_x(u(x))]$. Observe that $\nabla \ell(u) : x \mapsto \nabla \ell_x(u(x))$ is a gradient for $\ell$, and $\mathcal{L} = \ell \circ X$. Therefore, we can apply the variational bound to try to get a Kurdyka-Łojasiewicz property.

$$\|\nabla \mathcal{L}(\theta)\|_\Theta^2 = \sup_{\nu \in \Theta} \langle \nu, \nabla \mathcal{L}(\theta) \rangle_\Theta^2 / \|\nu\|_\Theta^2 = \sup_{\nu \in \Theta} \langle X(\nu), \nabla \ell(u) \rangle_\mathcal{D}^2 / \|\nu\|_\Theta^2$$

We can then evaluate at a well-chosen point ($\nu = \zeta \in \Theta$). For $\mathcal{D}$-almost all $x \in \mathcal{X}$, define $i^* = \arg\max_i \langle \zeta, x \rangle$, together with $M_x = \langle \zeta_{i^*}, x \rangle$ and $m_x = \max_{i \neq i^*} \langle \zeta_i, x \rangle$. By the $\varepsilon$-margin separability assumption, it holds $M_x \geq m_x + \varepsilon \|\zeta\|_\Theta$. Therefore, with the notation $p_{x,i} = E(u(x))_i$

$$\langle X(\zeta), y - p \rangle_\mathcal{D} = \mathbb{E}_x \left[ \sum_{i \in [c]} \langle \zeta_i, x \rangle (y_{x,i} - p_{x,i}) \right] = \mathbb{E}_x \left[ M_x(1 - p_{x,i^*}) - \sum_{i \neq i^*} \langle \zeta_i, x \rangle p_{x,i} \right] \quad (1)$$

$$\geq \mathbb{E}_x \left[ M_x(1 - p_{x,i^*}) - m_x \sum_{i \neq i^*} p_{x,i} \right] = \mathbb{E}_x \left[ (M_x - m_x)(1 - p_{x,i^*}) \right]$$

$$\geq \varepsilon \|\zeta\|_\Theta \mathbb{E}_x \left[ 1 - p_{i^*} \right] = \varepsilon \|\zeta\|_\Theta \mathbb{E}_x \left[ 1 - e^{-\ell_x(u(x))} \right] \geq \varepsilon \kappa \|\zeta\|_\Theta \left( 1 - e^{-\ell(u)} \right) \quad (2)$$

where (1) is because $(\inf_\lambda \mathcal{L}(\lambda \zeta) = 0)$ implies $y_{x,i} = \delta_{i=i^*}$ (see appendix A.5.2), and (2) is by $\varepsilon$-margin separability assumption then definition of $\ell_x$ and finally $\mathbb{E}[\psi(Z)] \geq \mathbb{P}(Z \geq \mathbb{E}[Z]) \psi(\mathbb{E}[Z])$ for any non-negative random variable $Z$ since $\psi : z \in \mathbb{R}_+ \mapsto 1 - e^{-z}$ is increasing and non-negative. The final result follows from $\mathrm{d}\varphi_{\mathcal{L}(\theta)} \|\nabla \mathcal{L}(\theta)\|_\Theta^2 \geq \varepsilon^2 \kappa^2$, by integration of $\mathrm{d}\varphi_z = (1 - e^{-z})^{-2}$ to get $\varphi : z \in \mathbb{R}_+^* \mapsto \log(e^{+z} - 1) - (e^{+z} - 1)^{-1}$ and thus $\varphi^{-1} : u \in \mathbb{R} \mapsto \log(1 + 1/W_0(e^{-u}))$.

### A.5.4 Logistic bound asymptotic behavior

We show here that the convergence bound for the logistic regression presented in Proposition 4.5 is consistent with the previously-known asymptotic $\mathcal{O}(1/t)$ behavior.

Let $(C, \tau) \in \mathbb{R} \times \mathbb{R}_+^*$ and $f : t \mapsto \log\left(1 + \frac{1}{W_0(\exp(t/\tau - C))}\right)$. Let us show $f(t) \underset{+\infty}{=} \mathcal{O}(1/t)$.

As warmup, note that $\exp(t/\tau - C) \underset{t \to +\infty}{\longrightarrow} +\infty$, and $W_0(x) \underset{x \to +\infty}{\longrightarrow} +\infty$, therefore $f(t) \underset{t \to +\infty}{\longrightarrow} 0$.

From Hoorfar and Hassani [2008, Theorem 2.1], for $x \geq e$ it holds $W_0(x) \geq \log(x) - \log(\log(x))$. Therefore, for $t$ sufficiently large, it holds

$$\log\left(1 + \frac{1}{W_0(\exp(t/\tau - C))}\right) \leq \frac{1}{W_0(\exp(t/\tau - C))} \leq \frac{1}{t/\tau - C - \log(t/\tau - C)} = \mathcal{O}(1/t)$$

### A.5.5 Discussion of assumptions for the logistic bound

**Separation assumption.** The existence of an $\varepsilon$-separating ray for some $\varepsilon > 0$ in Proposition 4.5 is identical to the separation assumption Soudry et al. [2018, Assumption 4] (multi-class version, which itself recovers Soudry et al. [2018, Assumption 1] in the two-class case, which is the standard notion of "linear separability"). Then $\inf_\lambda \mathcal{L}(\lambda\zeta) = 0$ is consistency of the ray $\zeta$ with the class labels.

Indeed, the linear separability assumption is that for a dataset $(x_i, y_i) \in \mathbb{R}^d \times [c]$ for $i \in [n]$, there exists a vector $w \in \mathbb{R}^{c \times d}$ such that for all $i \in [n]$, and for all $k \in [c]$, if $k \neq y_i$, then $w_k \cdot x_i - w_{y_i} \cdot x_i < 0$. Let $\varepsilon = \inf_i \inf_{k \neq y_i} -(w_k \cdot x_i - w_{y_i} \cdot x_i)$. Since the number of training points is finite and the number of classes is finite, this infimum is a minimum, and thus $\varepsilon > 0$. It follows immediately that $w$ is an $(\varepsilon/\|w\|_2)$-separating ray, and satisfies $\inf_\lambda \mathcal{L}(\lambda w) = 0$.

The difference is only that our assumption is quantified, because $\varepsilon$ appears explicitly in our bound, whereas it was previously abstracted away by the Landau asymptotic notation. To properly quantify this notion of separation margin, one must be careful with the fact that the unquantified separation assumption is invariant by positive rescaling of the separating vector. We have chosen to define the ray $\zeta$ only up to a positive constant, whereas in [Soudry et al., 2018], a cancellation of the norm of the separating vector is chosen instead (convergence to $w^*/\|w^*\|$), but the two viewpoints are equivalent.

**Isolation assumption.** Previous works operating in the finite-data regime did not explicitly have a mention of an isolation assumption. Indeed, for a finitely supported distribution $p$, one can simply take $\kappa = \min_i p_i$, as noted in Section 4.3. For a dataset of size $n$ with equally-weighted samples, this reduces to $\kappa = 1/n$ and can again be abstracted away in asymptotic notation. Since we have chosen to give explicit bounds, we must make that constant appear, hence the existence of the assumption.

We could have used $1/n$ in place of the introduction of the notion of isolation, but this would have forced a vacuous bound in the infinite-data regime, whereas a positive isolation constant guarantees convergence even with continuous distributions. We try hereafter to give a better intuition of why such a positive isolation might be proven in typical machine learning scenarii.

The use of $\kappa$ in the proof is $\mathbb{E}[\psi(Y)] \geq \kappa \psi(\mathbb{E}[Y])$ when $Y$ is $\kappa$-isolated and $\psi$ increasing. This is because we measure only the average loss, the pointwise loss averaged over points in the dataset, which could be driven by the loss on a single point. This happens precisely when there remains exactly one misclassified point $i_0$, while other points are correctly classified, i.e. $\ell = \frac{1}{n}\sum_i \ell_i \approx \frac{1}{n}\ell_{i_0}$. This local misclassification is possible because there is 1 point which is sufficiently "isolated", hence the $1/n$, however if the dataset came with point-pairs very close to each other and identical labels, then it would become essentially impossible for a sufficently regular classifier to misclassify exactly one point, leading to a factor of $2/n$ instead (the corresponding amount of mass "isolated").
For a fixed number of training points $n \in \mathbb{N}$, there always exists a dataset with a single isolated point, thus the bound $\kappa \geq 1/n$ is tight without assumptions on the data generation process. However, there is typically an assumption in machine learning that we have not leveraged here: as the size of the dataset increases, the distribution of the data does not change, for instance all samples are taken independently identically distributed with a fixed distribution. Thus, $\kappa$ need not vanish as $n \to +\infty$. The regularity of the underlying distribution and the regularity of the classifier (obtained by finiteness of $\|\theta\|_2$) could be analyzed together to derive a positive limit for $\kappa$. Should a proof for such a property become available in the future, it could be chained with Proposition 4.5 as-is directly to obtain a better convergence speed. The use of $\kappa$ rather than $1/n$ in our bound is meant to highlight this possibility explicitly. We otherwise use $\kappa = 1/n$ in experiments.

### A.5.6 Experiments for logistic regression

The following figures show examples of the convergence speed observed with gradient descent and step size 0.1 in different scenarios. In Fig. 5 we depict a configuration where the bound we presented in Proposition 4.5 accurately describes the observed evolution of the loss, including the flat startup, the sudden drop and its position, and the asymptotic regime $\mathcal{L}(\theta_t) \leq \frac{1}{\alpha + \beta t}$. In Fig. 6, we depict a more realistic configuration, where the general behavior observed is similar, but the bound's constants are off by several orders of magnitude. In both cases, we take as isolation constant $\kappa = 1/n$.

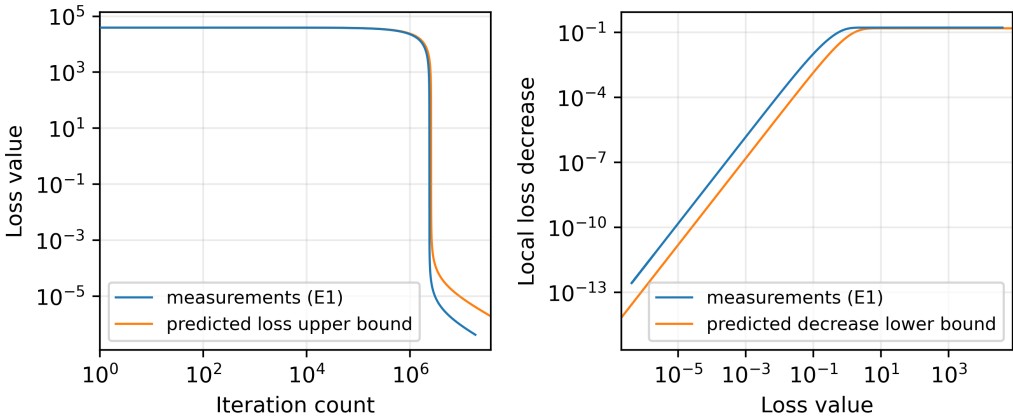

Figure 5: Logistic regression on $n = 3$ samples in dimension $d = 4$ with $c = 3$ classes. The data is hand-picked to show a tight regime of the bound. Measurements and predicted curves overlap at first.

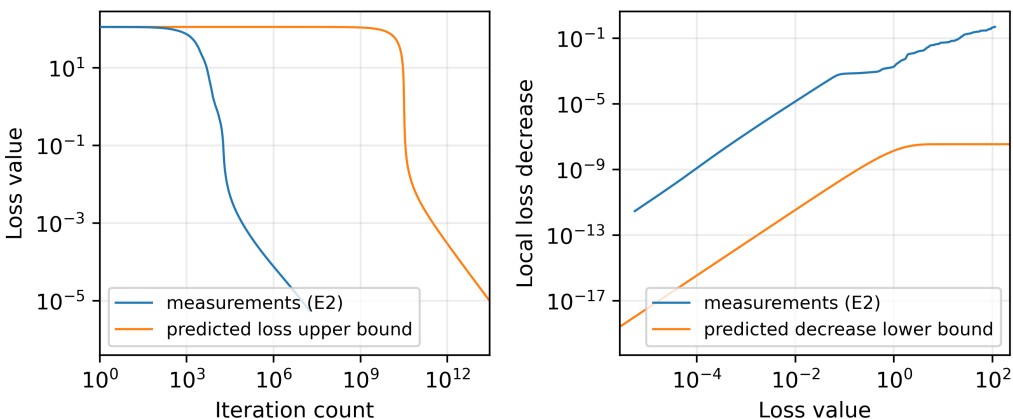

Figure 6: Logistic regression on $n = 100$ samples in dimension $d = 5$ with $c = 4$ classes. The data points, optimal direction and initial point are drawn at random from gaussian distributions.

These two experiments were conducted in parallel on an Intel i7 CPU, for a total running time of 14h.

### A.6  Periodic signal recovery, paired subcase

For shortness in the following proof, for any $\omega \in \mathbb{R}$, let $e_\omega \in \mathcal{F}$ be the function $e_\omega : x \mapsto \sin(\omega x)$, and $e'_\omega = \frac{\partial}{\partial \omega} e_\omega \in \mathcal{F}$ its derivative, $e'_\omega : x \mapsto x \cos(\omega x)$. Moreover, in all the following, we let $\psi = -\operatorname{sinc}'$ and $\phi = -\operatorname{sinc}''$ to avoid writing apostrophes and additional negative signs everywhere. Recall that we let $x_0$ be the first zero of $x \mapsto \phi(x)$, that is to say $x_0 \approx 2.0815$.

*Proof of Proposition 4.7.* Let $\varepsilon = \eta/R$ and $\delta = \mu/R$. Note that it holds $R\varepsilon \leq x_0$ and $\varepsilon < \frac{1}{2}\mu$. Let $\theta = (a, \omega) \in \Theta$ such that $(\omega, \omega^*)$ is $\frac{\eta}{R}$-paired. Let $h = F(a, \omega) - f^*$. By the assumption $\ell(F(a, \omega)) \neq 0$, we know $\|h\|_\mathcal{D} \neq 0$. We will show that $\mathrm{R}\left(K_\theta^\star; h, h\right)$ is bounded below by some constant. We defer the proof that this constant is positive (non-degeneracy of the bound) to a later section.

Let $g_{0,k} = e_{\omega_k}$ and $g_{1,k} = e_{\omega_k^*} - e_{\omega_k}$. Observe that $h \in \operatorname{Span}(g)$ because

$$h = \sum_k a_k e_{\omega_k} - a_k^* e_{\omega_k^*} = \sum_k (a_k - a_k^*) e_{\omega_k} + a_k^* (e_{\omega_k} - e_{\omega_k^*})$$

Let $b_{0,k} = (\delta_{i=k}, 0)_{i\in[m]} \in \Theta$, and $b_{1,k} = \left(0, \delta_{i=k}\frac{1}{a_k}(\omega_k - \omega_k^*)\right)_{i\in[m]} \in \Theta$, so that it holds $\mathrm{d}F_\theta \cdot b_{0,k} = e_{\omega_k}$ and $\mathrm{d}F_\theta \cdot b_{1,k} = (\omega_k - \omega_k^*)\, e'_{\omega_k}$.

$$\mathrm{R}\left(K_\theta^\star; h, h\right) = \sup_{\nu \in \Theta} \mathrm{R}\left(\mathrm{d}F_\theta^\star; \nu, h\right)^2 \tag{1}$$

$$\geq \frac{\lambda_{\min}(\mathrm{R}\left(\langle\cdot,\cdot\rangle_{\mathcal{D}}; \mathrm{d}F_\theta \cdot b_i, g_j\right)_{i,j})^2 \min_{(i,u)\in[m]\times[2]}\|\mathrm{d}F_\theta \cdot b_{i,u}\|_{\mathcal{D}}^2/\|b_{i,u}\|_\Theta^2}{\lambda_{\max}\left(\mathrm{R}\left(\langle\cdot,\cdot\rangle_\Theta; b_i, b_j\right)_{i,j}\right)\lambda_{\max}\left(\mathrm{R}\left(\langle\cdot,\cdot\rangle_{\mathcal{D}}; g_i, g_j\right)_{i,j}\right)} \tag{2}$$

where (1) is Proposition 3.4, and (2) is Proposition 3.6. In the above expression, the indices $(k, 0) \in ([m] \times \{0\})$ such that $a_k = a_k^*$ and the indices $(k, 1) \in ([m] \times \{1\})$ such that $\omega_k = \omega_k^*$ have been omitted (since the corresponding derivative is zero), and thus the matrices are all well-defined. For simplicity in the notation, and since the correction would just amount to selecting the corresponding subsets of $b$ and $g$ without altering the final result, we will just assume that $\forall k, a_k \neq a_k^*$ and $\forall k, \omega_k \neq \omega_k^*$ in the following, so the index set remains $([m] \times [2])$.

The first factor in the denominator is the largest eigenvalue of the identity, i.e. one. The second factor of the numerator, corresponding to the singular value, is $\min_{(i,u)}\|\mathrm{d}F_\theta \cdot b_{i,u}\|_{\mathcal{D}}^2/\|b_{i,u}\|_\Theta = \min_k(\|e_{\omega_k}\|_{\mathcal{D}}^2, a_k^2\|e'_{\omega_k}\|_{\mathcal{D}}^2) \geq \min(1, a_k^2)\left(\phi(R\varepsilon) - 1/(R(\delta - \varepsilon))\right) \geq \alpha(\phi(\eta) - 1/(\mu - \eta))$ (see Lemma A.5 for the lower bounds on the seminorms).

There remains only two matrices whose eigenvalues we need to bound.

We will proceed using Gershgorin's discs theorem [Gerschgorin, 1931] for the control of eigenvalues:

$$\forall X \in \mathbb{R}^{k\times k}, \ \lambda_{\min}(X) \geq \inf_{i\in[k]} X_{i,i} - \frac{1}{2}\sum_{j\neq i}|X_{i,j}| + |X_{j,i}|$$

$$\forall X \in \mathbb{R}^{k\times k}, \ \lambda_{\max}(X) \leq \sup_{i\in[k]} X_{i,i} + \frac{1}{2}\sum_{j\neq i}|X_{i,j}| + |X_{j,i}|$$

Starting with the denominator $z_0 = \lambda_{\max}(\mathrm{R}(\langle\cdot,\cdot\rangle_{\mathcal{D}}; g_i, g_j)_{i,j})$,

$$z_0 = \lambda_{\max}(\mathrm{R}(\langle\cdot,\cdot\rangle_{\mathcal{D}}; g_i, g_j) \leq \sup_{(i,u)\in[m]\times[2]} 1 + \sum_{(j,v)\neq(i,u)}|\mathrm{R}(\langle\cdot,\cdot\rangle_{\mathcal{D}}; g_i, g_j)| \tag{1}$$

$$\leq \sup_{i\in[m]} 1 + \frac{\psi(R\varepsilon) + \frac{1}{R(\delta-\varepsilon)}}{\phi(R\varepsilon) - \frac{1}{R(\delta-\varepsilon)}} + 4\sum_{j\neq i}\frac{\frac{1}{R(|j-i|\delta-2\varepsilon)} + \frac{1}{R((i+j+2)\delta-2\varepsilon)}}{\phi(R\varepsilon) - \frac{1}{R(\delta-\varepsilon)}} \tag{2}$$

$$\leq \sup_{i\in[m]} 1 + \frac{1}{\phi(R\varepsilon) - \frac{1}{R(\delta-\varepsilon)}}\left(\psi(R\varepsilon) + \frac{1}{R(\delta-\varepsilon)} + \frac{4}{R(\delta-2\varepsilon)}\sum_{j\neq i}\left(\frac{1}{|j-i|} + \frac{1}{i+j+2}\right)\right) \tag{3}$$

$$\leq 1 + \frac{1}{\phi(R\varepsilon) - \frac{1}{R(\delta-\varepsilon)}}\left(\psi(R\varepsilon) + \frac{1}{R(\delta-\varepsilon)} + \frac{4}{R(\delta-2\varepsilon)}4H_m\right) = 1 + \rho_0 \tag{4}$$

where (1) is Gershgorin's disc upper bound with a symmetric matrix, (2) is Proposition A.3 proven below, (3) is factorization of the terms depending on $(i,j)$ in the denominators (because if $j \neq i$, then $|j - i| \geq 1$), and (4) is $\sum_{j\neq i}\frac{1}{|j-i|} + \frac{1}{j+i+2} \leq \sum_{j<i}\frac{2}{|j-i|} + \sum_{j>i}\frac{2}{|j-i|} \leq 2\sum_{k=1}^m\frac{2}{k} = 4H_m$.

For the numerator $z_1 = \lambda_{\min}(\mathrm{R}(\mathrm{d}F_\theta^\star; b_i, g_j)$ now, we start by simplifying each entry

$$\mathrm{R}(\mathrm{d}F_\theta; b_i, g_j) = \frac{\langle \mathrm{d}F_\theta \cdot b_i, g_j\rangle_{\mathcal{D}}}{\|b_i\|_\Theta\|g_j\|_{\mathcal{D}}} = \frac{\langle \mathrm{d}F_\theta \cdot b_i, g_j\rangle_{\mathcal{D}}}{\|b_i\|_\Theta\|g_j\|_{\mathcal{D}}} = \mathrm{R}(\langle\cdot,\cdot\rangle_{\mathcal{D}}; h_i, g_j)$$

Then apply Gershgorin's disc lower bound

$$z_1 = \lambda_{\min}(\mathrm{R}(\,\mathrm{d}F_\theta^\star; b_i, g_j)_{i,j}) = \lambda_{\min}(\mathrm{R}(\langle\cdot,\cdot\rangle_{\mathcal{D}}; h_i, g_j)_{i,j})$$

$$\geq \inf_{(i,u)\in[m]\times[2]} \mathrm{R}(\langle\cdot,\cdot\rangle_{\mathcal{D}}; h_{i,u}, g_{i,u}) - \frac{1}{2} \sum_{(j,v)\neq(i,u)} |\mathrm{R}(\langle\cdot,\cdot\rangle_{\mathcal{D}}; h_{i,u}, g_{i,v})| + |\mathrm{R}(\langle\cdot,\cdot\rangle_{\mathcal{D}}; h_{j,v}, g_{i,u})|$$

$$\geq \inf_{i\in[m]} \frac{\phi(R\varepsilon) - \frac{1}{R(\delta-\varepsilon)}}{\phi(0) + \frac{1}{R(\delta-\varepsilon)}} - \left( \frac{\psi(R\varepsilon) + \frac{1}{R(\delta-\varepsilon)}}{\phi(R\varepsilon) - \frac{1}{R(\delta-\varepsilon)}} + 4 \sum_{j\neq i} \frac{\frac{1}{R(|j-i|\delta-2\varepsilon)} + \frac{1}{R((i+j+2)\delta-2\varepsilon)}}{\phi(R\varepsilon) - \frac{1}{R(\delta-\varepsilon)}} \right) \quad (1)$$

$$\geq \inf_{i\in[m]} \frac{\phi(R\varepsilon) - \frac{1}{R(\delta-\varepsilon)}}{\phi(0) + \frac{1}{R(\delta-\varepsilon)}} - \rho_0 = \kappa_0 - \rho_0 \quad (2)$$

where (1) is Proposition A.2 for the left term and Proposition A.3 for the right term (with a symmetric upper-bound), and (2) is the same upper-bound for off-diagonal terms as calculated above. $\qquad\square$

*Proof of non-degeneracy for Proposition 4.7.* Recall the definition of the constants

$$\kappa_0(\mu,\eta) = \frac{\phi(\eta) - \frac{1}{\mu-\eta}}{\phi(0) + \frac{1}{\mu-\eta}} \qquad \rho_0(\mu,\eta) = \frac{\psi(\eta) + \frac{1}{\mu-\eta} + \frac{4H}{\mu-2\eta}}{\phi(\eta) - \frac{1}{\mu-\eta}}$$

By continuity, to show $\exists \eta > 0$ s.t. $\kappa_0(\mu,\eta) > \rho_0(\mu,\eta)$, it is sufficient to show $\kappa_0(\mu,0) > \rho_0(\mu,0)$. These values are $\kappa_0(\mu,0) = \left(\phi(0) - \frac{1}{\mu}\right) / \left(\phi(0) + \frac{1}{\mu}\right)$ and $\rho_0(\mu,0) = \left(\frac{1}{\mu} + \frac{4H}{\mu}\right) / \left(\phi(0) - \frac{1}{\mu}\right)$.

Using $\phi(0) = 1/3$ and reorganizing terms, the equation $\kappa_0(\mu,0) > \rho_0(\mu,0)$ is satisfied if and only if

$$\left(\frac{\mu}{3} - 1\right)^2 - (1 + 4H)\left(\frac{\mu}{3} + 1\right) > 0$$

This is a polynomial in $\mu$ of degree two, and positive at infinity, therefore letting $\mu_0$ be its largest root, it holds for all $\mu > \mu_0$ that $\kappa_0(\mu,0) > \rho_0(\mu,0)$. $\qquad\square$

**Proposition A.2** (Auxiliary for on-diagonal control)**.** *If $(\omega, \omega^*)$ is $\varepsilon$-paired and $\omega^*$ is $\delta$-separated, where it holds $R\varepsilon \leq x_0$, and $\varepsilon < \frac{1}{2}\delta$, and $\forall i, \omega_i \neq \omega_i^*$, then*

$$\forall(i,u) \in [m]\times[2], \ \mathrm{R}(\langle\cdot,\cdot\rangle_{\mathcal{D}}; g_{i,u}, h_{i,u}) \geq \frac{\phi(R\varepsilon) - \frac{1}{R(\delta-\varepsilon)}}{\phi(0) + \frac{1}{R(\delta-\varepsilon)}}$$

*where $g, h \in \mathcal{F}^{m\times 2}$ satisfy $g_{i,0} = h_{i,0} = e_{\omega_i}$ and $g_{i,1}, h_{i,1} \in \{e_{\omega_k} - e_{\omega_k^*}, (\omega_k - \omega_k^*)e_{\omega_k}'\}$.*

**Proposition A.3** (Auxiliary for off-diagonal control)**.** *If $(\omega, \omega^*)$ is $\varepsilon$-paired and $\omega^*$ is $\delta$-separated and ordered $(i \leq j \Rightarrow \omega_i^* \leq \omega_j^*)$, where it holds $R\varepsilon \leq x_0$, and $\varepsilon < \frac{1}{2}\delta$, and $\forall i, \omega_i \neq \omega_i^*$, then*

$$\forall i \in [m], \quad \sup_{\substack{(u,v)\in[2]\times[2] \\ u\neq v}} |\mathrm{R}(\langle\cdot,\cdot\rangle_{\mathcal{D}}; g_{i,u}, h_{i,v})| \leq \frac{\psi(R\varepsilon) + \frac{1}{R(\delta-\varepsilon)}}{\phi(R\varepsilon) - \frac{1}{R(\delta-\varepsilon)}}$$

$$\forall(i,j) \in [m]\times[m], \ i\neq j \Rightarrow \sup_{(u,v)\in[2]\times[2]} |\mathrm{R}(\langle\cdot,\cdot\rangle_{\mathcal{D}}; g_{i,u}, h_{j,v})| \leq \frac{\frac{1}{R(|j-i|\delta-2\varepsilon)} + \frac{1}{R((i+j+2)\delta-2\varepsilon)}}{\phi(R\varepsilon) - \frac{1}{R(\delta-\varepsilon)}}$$

*where $g, h \in \mathcal{F}^{m\times 2}$ satisfy $g_{i,0} = h_{i,0} = e_{\omega_i}$ and $g_{i,1}, h_{i,1} \in \{e_{\omega_k} - e_{\omega_k^*}, (\omega_k - \omega_k^*)e_{\omega_k}'\}$.*

The proof for both propositions is a case disjunction. We will state an intermediate lemma first.

**Lemma A.4** (Cosine bound by cross-ratio control)**.**
*Let $a : [0,1] \to \mathcal{F}$ and $b : [0,1] \to \mathcal{F}$. If there exists $\alpha \in [0,1]$ such that it holds*

$$\forall(s,t,u,v) \in [0,1]^4, \ \frac{\langle a(s), b(u)\rangle_{\mathcal{D}} \langle a(t), b(v)\rangle_{\mathcal{D}}}{\langle a(s), a(t)\rangle_{\mathcal{D}} \langle b(u), b(v)\rangle_{\mathcal{D}}} \leq \alpha^2$$

*then $\left| \mathrm{R}\left(\langle\cdot,\cdot\rangle_{\mathcal{D}}; \int_0^1 a(s)\,\mathrm{d}s, \int_0^1 b(t)\,\mathrm{d}t\right) \right| \leq \alpha$.*

*Proof.* By expanding the definition (1), bilinearity (2), then applying the hypothesis pointwise (3).

$$R\left(\langle\cdot,\cdot\rangle_{\mathcal{D}};\int_0^1 a(s)\,\mathrm{d}s,\int_0^1 b(t)\,\mathrm{d}t\right)^2 = \frac{\left\langle\int_0^1 a(s)\,\mathrm{d}s,\int_0^1 b(t)\,\mathrm{d}t\right\rangle_{\mathcal{D}}\left\langle\int_0^1 a(u)\,\mathrm{d}u,\int_0^1 b(v)\,\mathrm{d}v\right\rangle_{\mathcal{D}}}{\left\langle\int_0^1 a(s)\,\mathrm{d}s,\int_0^1 a(u)\,\mathrm{d}u\right\rangle_{\mathcal{D}}\left\langle\int_0^1 b(t)\,\mathrm{d}t,\int_0^1 b(v)\,\mathrm{d}v\right\rangle_{\mathcal{D}}} \quad (1)$$

$$= \frac{\int_0^1\int_0^1\int_0^1\int_0^1\langle a(s),b(t)\rangle_{\mathcal{D}}\langle a(u),b(v)\rangle_{\mathcal{D}}\,\mathrm{d}s\,\mathrm{d}t\,\mathrm{d}u\,\mathrm{d}v}{\int_0^1\int_0^1\int_0^1\int_0^1\langle a(s),a(u)\rangle_{\mathcal{D}}\langle b(t),b(v)\rangle_{\mathcal{D}}\,\mathrm{d}s\,\mathrm{d}t\,\mathrm{d}u\,\mathrm{d}v} \quad (2)$$

$$\leq \alpha^2 \quad (3)$$

An upper bound on the cross-ratio allows interversions under the integral, thus the result. $\qquad\square$

*Proof of Proposition A.3.* For shortness, let $I_i = \{(1-t)\omega_i + t(\omega_i^*), t\in[0,1]\} \subseteq \mathbb{R}$. By observing that $e_{\omega_k} - e_{\omega_k^*} = \int_0^1 e'_{q(t)}(\omega_k - \omega_k^*)\,\mathrm{d}t$ for $q(t) = (1-t)\omega_k^* + t\omega_k$ on one hand, and $(\omega_k - \omega_k^*)e'_{\omega_k} = \int_0^1 e'_{r(t)}(\omega_k - \omega_k^*)\,\mathrm{d}t$ for $r(t) = \omega_k$ on the other hand, we reduce to cross-ratio upper bounds only.

For the first part of the proof, let $i \in [m]$. By symmetry, it is sufficient to consider $(u=0, v=1)$.

$$\forall p,q \in I_i,\ \frac{\langle e'_p, e_{\omega_i}\rangle_{\mathcal{D}}\langle e'_q, e_{\omega_i}\rangle_{\mathcal{D}}}{\langle e'_p, e'_q\rangle_{\mathcal{D}}\langle e_{\omega_i}, e_{\omega_i}\rangle_{\mathcal{D}}} \leq \frac{\left(\psi(R\varepsilon)+\frac{1}{R(\delta-\varepsilon)}\right)^2}{\left(1-\frac{1}{R(\delta-\varepsilon)}\right)\left(\phi(R\varepsilon)-\frac{1}{R(\delta-\varepsilon)}\right)} \leq \left(\frac{\psi(R\varepsilon)-\frac{1}{R(\delta-\varepsilon)}}{\phi(R\varepsilon)-\frac{1}{R(\delta-\varepsilon)}}\right)^2$$

by Lemma A.5, where the last step is $\phi(R\varepsilon) \leq \phi(0) = 1/3 \leq 1$. Conclude by Lemma A.4.

For the second part of the proposition, let $(i,j) \in [m] \times [m]$, and proceed by case disjunction on $(u,v) \in [2] \times [2]$. Let $(p_i, q_i) \in I_i \times I_i$ and $(p_j, q_j) \in I_j \times I_j$, and observe that

$$(u=0,v=0)\quad \frac{\langle e_{\omega_i}, e_{\omega_j}\rangle_{\mathcal{D}}\langle e_{\omega_i}, e_{\omega_j}\rangle_{\mathcal{D}}}{\langle e_{\omega_i}, e_{\omega_i}\rangle_{\mathcal{D}}\langle e_{\omega_j}, e_{\omega_j}\rangle_{\mathcal{D}}} \leq \frac{\left(\frac{1}{R(|j-i|\delta-2\varepsilon)}+\frac{1}{R((i+j+2)\delta-2\varepsilon)}\right)^2}{\left(1-\frac{1}{R(\delta-\varepsilon)}\right)^2}$$

$$(u=0,v=1)\quad \frac{\langle e_{\omega_i}, e'_{p_j}\rangle_{\mathcal{D}}\langle e_{\omega_i}, e'_{q_j}\rangle_{\mathcal{D}}}{\langle e_{\omega_i}, e_{\omega_i}\rangle_{\mathcal{D}}\langle e'_{p_j}, e'_{q_j}\rangle_{\mathcal{D}}} \leq \frac{\left(\frac{1}{R(|j-i|\delta-2\varepsilon)}+\frac{1}{R((i+j+2)\delta-2\varepsilon)}\right)^2}{\left(1-\frac{1}{R(\delta-\varepsilon)}\right)\left(\phi(R\varepsilon)-\frac{1}{R(\delta-\varepsilon)}\right)}$$

$$(u=1,v=1)\quad \frac{\langle e'_{p_i}, e'_{p_j}\rangle_{\mathcal{D}}\langle e'_{q_i}, e'_{q_j}\rangle_{\mathcal{D}}}{\langle e'_{p_i}, e'_{q_i}\rangle_{\mathcal{D}}\langle e'_{p_j}, e'_{q_j}\rangle_{\mathcal{D}}} \leq \frac{\left(\frac{1}{R(|j-i|\delta-2\varepsilon)}+\frac{1}{R((i+j+2)\delta-2\varepsilon)}\right)^2}{\left(\phi(R\varepsilon)-\frac{1}{R(\delta-\varepsilon)}\right)^2}$$

by Lemma A.5. The case $(u=1, v=0)$ is identical to $(u=0, v=1)$ by symmetry, and conclusion follows as above by $\phi(R\varepsilon) \leq 1$ then Lemma A.4. $\qquad\square$

*Proof of Proposition A.2.* Let $i \in [m]$. We will prove $R(\langle\cdot,\cdot\rangle_{\mathcal{D}}; g_{i,u}, h_{i,u}) \geq \kappa_0$ by case disjunction on $u \in [2]$. For the case $u=0$, observe that $R(\langle\cdot,\cdot\rangle_{\mathcal{D}}; e_{\omega_k}, e_{\omega_k}) = 1$. Since $\phi$ is decreasing on $[0, R\varepsilon]$ (see A.6.2), and $\kappa_0 \leq 1$, the conclusion is immediate.

For the case $u=1$, identically to the proof of Prop A.3 above, let $(p,q,r,s) \in I_i^4$.

$$\frac{\langle e'_p, e'_q\rangle_{\mathcal{D}}\langle e'_r, e'_s\rangle_{\mathcal{D}}}{\langle e'_p, e'_r\rangle_{\mathcal{D}}\langle e'_q, e'_s\rangle_{\mathcal{D}}} \geq \left(\frac{\phi(R\varepsilon)-\frac{1}{R(\delta-\varepsilon)}}{\phi(0)-\frac{1}{R(\delta-\varepsilon)}}\right)^2 = \kappa_0^2$$

by Lemma A.5. Thus by expanding integrals as for Lemma A.4, the cross-ratio lower bound implies

$$R\left(\langle\cdot,\cdot\rangle_{\mathcal{D}};\int g,\int h\right) = \frac{\langle\int g,\int h\rangle_{\mathcal{D}}}{\sqrt{\langle\int g,\int g\rangle_{\mathcal{D}}\langle\int h,\int h\rangle_{\mathcal{D}}}} \geq \kappa_0$$

$\qquad\square$

**Lemma A.5.** *If $(\omega, \omega^*)$ is $\varepsilon$-paired and $\omega^*$ is $\delta$-separated and ordered, $R\varepsilon \leq x_0$ and $\varepsilon < \frac{1}{2}\delta$, then*

$$\forall i, \quad \forall u \in I_i, \qquad\qquad \langle e_u, e_u \rangle_{\mathcal{D}} \geq \frac{1}{2} - \frac{1}{2R(\delta - \varepsilon)}$$

$$\forall i, \quad \forall (u,v) \in I_i \times I_i, \quad \frac{1}{R}|\langle e_u', e_v \rangle_{\mathcal{D}}| \leq \frac{\psi(R\varepsilon)}{2} + \frac{1}{2R(\delta - \varepsilon)}$$

$$\forall i, \quad \forall (u,v) \in I_i \times I_i, \quad \frac{1}{R^2}\langle e_u', e_v' \rangle_{\mathcal{D}} \in \left[ \frac{1}{2}\phi(R\varepsilon) - \frac{1}{2R(\delta - \varepsilon)}, \frac{1}{2}\phi(0) + \frac{1}{2R(\delta - \varepsilon)} \right]$$

*Additionally, if $i \neq j$, then for all $(u,v) \in I_i \times I_j$, it holds*

$$\max\left( |\langle e_u, e_v \rangle_{\mathcal{D}}|, \frac{1}{R}|\langle e_u', e_v \rangle_{\mathcal{D}}|, \frac{1}{R^2}|\langle e_u', e_v' \rangle_{\mathcal{D}}| \right) \leq \frac{1}{2R(|i-j|\delta - 2\varepsilon)} + \frac{1}{2R((i+j+2)\delta - 2\varepsilon)}$$

*where $I_i = \{(1-t)\omega_i + t\omega_i^*, t \in [0,1]\} \subseteq \mathbb{R}$.*

*Proof.* The idea is to first compute dot products in closed forms to make the cardinal sine function (sinc : $x \mapsto \sin(x)/x$) appear, then rely on properties of the cardinal sine and its derivatives to prove each property by case disjunction. Therefore, for any $(u,v) \in \mathbb{R}_+ \times \mathbb{R}_+$, compute the integral (assuming $u \neq v$ and completing by continuity) using $[2\sin(a)\sin(b) = \cos(a-b) - \cos(a+b)]$,

$$\begin{aligned}
\langle e_u, e_v \rangle_{\mathcal{D}} &= \frac{1}{2R} \int_{-R}^{+R} \sin(ux)\sin(vx)\,\mathrm{d}x \\
&= \frac{1}{4R} \int_{-R}^{+R} \cos((u-v)x) - \cos((u+v)x) \\
&= \frac{1}{4R} \left[ \frac{\sin((u-v)x)}{u-v} - \frac{\sin((u+v)x)}{u+v} \right]_{-R}^{+R} \\
&= \frac{1}{2}\left( \mathrm{sinc}(Ru - Rv) - \mathrm{sinc}(Ru + Rv) \right)
\end{aligned}$$

Compute the others by derivation

$$\begin{aligned}
\langle e_u', e_v \rangle_{\mathcal{D}} &= \frac{\partial}{\partial u}\langle e_u, e_v \rangle_{\mathcal{D}} = \frac{R}{2}\left( \mathrm{sinc}'(Ru - Rv) - \mathrm{sinc}'(Ru + Rv) \right) \\
\langle e_u', e_v' \rangle_{\mathcal{D}} &= \frac{\partial}{\partial u}\frac{\partial}{\partial v}\langle e_u, e_v \rangle_{\mathcal{D}} = \frac{R^2}{2}\left( -\mathrm{sinc}''(Ru - Rv) - \mathrm{sinc}''(Ru + Rv) \right)
\end{aligned}$$

The proof of all statements will then follow from a couple of properties of sinc and its derivatives:

1. $\forall x \in \mathbb{R}, \max\left\{ |\mathrm{sinc}(x)|, |\mathrm{sinc}'(x)|, |\mathrm{sinc}''(x)| \right\} \leq \frac{2}{|x|}$

2. $(-\mathrm{sinc}'')$ is non-negative decreasing on $[0, x_0]$, where $x_0 \approx 2.0815$ is its first zero.

3. $(-\mathrm{sinc}')$ is non-negative increasing on $[0, x_0]$.

These properties are depicted in Figure 7, and proven in Appendix A.6.2.

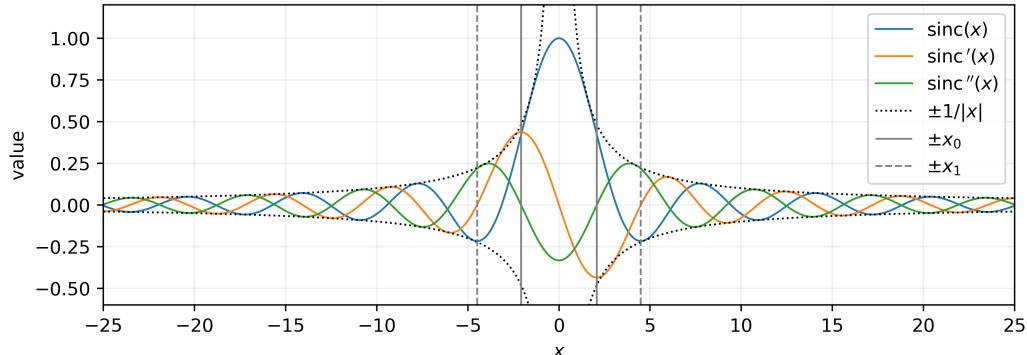

Figure 7: sinc and derivatives, first zeros $\operatorname{sinc}'(x_1) = 0$, $\operatorname{sinc}''(x_0) = 0$, $\frac{1}{|x|}$ envelope on $\pm[x_1, +\infty[$

For the first property, let $i \in [m]$, let $u \in I_i$, and observe that

$$\langle e_u, e_u \rangle_{\mathcal{D}} = \frac{1}{2} \left( \operatorname{sinc}(0) - \operatorname{sinc}(2Ru) \right) \geq \frac{1}{2} \left( 1 - \frac{2}{2R|u|} \right) \geq \frac{1}{2} - \frac{1}{2R(\delta - \varepsilon)}$$

since $|u| \geq |\omega_i^*| - |\omega_i - \omega_i^*| \geq \delta - \varepsilon$.

For the second property, let $i \in [m]$, and $(u, v) \in I_i \times I_i$. Without loss of generality, assume $u \leq v$,

$$\frac{1}{R} \left| \langle e'_u, e'_v \rangle_{\mathcal{D}} \right| = \frac{1}{R} \left| \frac{R}{2} \left( \operatorname{sinc}'(Ru - Rv) - \operatorname{sinc}'(Ru + Rv) \right) \right|$$

$$\leq \frac{1}{2} \left( \left| \operatorname{sinc}'(Ru - Rv) \right| + \left| \operatorname{sinc}'(Ru + Rv) \right| \right)$$

$$\leq \frac{1}{2} \left( \psi(R\varepsilon) + \frac{2}{R|u + v|} \right)$$

$$\leq \frac{\psi(R\varepsilon)}{2} + \frac{1}{2R(\delta - \varepsilon)}$$

since $|\operatorname{sinc}'(Ru - Rv)| = -\operatorname{sinc}'(Ru - Rv) \leq -\operatorname{sinc}'(R\varepsilon)$ by increase of $\psi = -\operatorname{sinc}'$ and $|u - v| \leq \varepsilon$ for the first term, and $\min(|u|, |v|) \geq \delta - \varepsilon$ for the second term.

For the third property, let $i \in [m]$ and $(u, v) \in I_i \times I_i$, without loss of generality $u \leq v$.

$$\frac{1}{R^2} \langle e'_u, e'_v \rangle_{\mathcal{D}} = \frac{1}{R^2} \frac{R^2}{2} \left( -\operatorname{sinc}''(Ru - Rv) - \operatorname{sinc}''(Ru + Rv) \right)$$

$$\in \left[ \frac{1}{2} \phi(R\varepsilon) - \frac{1}{2R(\delta - \varepsilon)}, \frac{1}{2} \phi(0) + \frac{1}{2R(\delta - \varepsilon)} \right]$$

since $\phi(Ru - Rv) \in [\phi(R\varepsilon), \phi(0)]$ by decrease of $\phi$ and since $|u - v| \leq \varepsilon$ for the first term, and $\phi(Ru + Rv) \in [-1/(R(\delta - \varepsilon)), +1/(R(\delta - \varepsilon))]$ since $\min(|u|, |v|) \geq \delta - \varepsilon$ for the second term.

Finally, for the last property, let $(i, j) \in [m] \times [m]$ such that $i \neq j$, and $(u, v) \in I_i \times I_j$.

$$\max \left( |\langle e_u, e_v \rangle_{\mathcal{D}}|, \frac{1}{R} |\langle e'_u, e_v \rangle_{\mathcal{D}}|, \frac{1}{R^2} |\langle e'_u, e'_v \rangle_{\mathcal{D}}| \right) \leq \frac{1}{2} \left( \frac{1}{R(|i - j|\delta - 2\varepsilon)} + \frac{1}{R((i + j + 2)\delta - 2\varepsilon)} \right)$$

Because it holds $|u - v| \geq |j - i|\delta - 2\varepsilon$ and $|u| + |v| \geq (i + 1)\delta - \varepsilon + (j + 1)\delta - \varepsilon$. $\qquad \square$

### A.6.1 Summary of the periodic signal recovery convergence argument

The proof is a little involved and the computations hard to follow, but the interesting part is that the proof is broken down, by relatively easy steps, into smaller statements that can be checked independently of each other. First, by Prop 3.3, convergence proofs on the quadratic loss can be reduced to control of a Rayleigh quotient away from zero. Secondly, by Prop 3.6, Rayleigh

quotient control is reduced to some easy singular values computation and an eigenvalue control of a matrix of simpler Rayleigh quotients in a well-chosen basis. Thirdly, by Gershgorin's disc theorem, the eigenvalue control is reduced to a number of upper bounds on cosine similarities. Then, by Lemma A.4, the numerous upper bounds on cosine similarities are reduced to upper bounds on cross-ratios to reduce the number of distinct cases to consider. Finally, by Lemma A.5, each cross-ratio bound is reduced to the analysis of a real-valued function on a small interval.

### A.6.2    Properties of the cardinal sine and derivatives

By definition, $\mathrm{sinc}(x) = \sin(x)/x$, thus $|\mathrm{sinc}(x)| \leq 1/|x| \leq 2/|x|$. Moreover, we will show $\mathrm{sinc}(x) \leq 1$. By symmetry (since $\mathrm{sinc}(-x) = \mathrm{sinc}(x)$), it is sufficient to show that $\sin(x) \leq x$ for all $x \in \mathbb{R}_+$, which holds because $\rho : x \in \mathbb{R}_+ \mapsto x - \sin(x)$ satisfies $\rho(0) = 0$ and is decreasing because $\rho'(x) = 1 - \cos(x) \leq 0$.

By derivation of the quotient, $\mathrm{sinc}' : x \mapsto \frac{\cos(x)}{x} - \frac{\sin(x)}{x^2}$, thus by triangular inequality and the above,

$$\left|\mathrm{sinc}'(x)\right| = \left|\frac{\cos(x)}{x} - \frac{\mathrm{sinc}(x)}{x}\right| \leq \frac{|\cos(x)|}{|x|} + \frac{|\mathrm{sinc}(x)|}{|x|} \leq \frac{2}{|x|}$$

Now let us show that $|\mathrm{sinc}'(x)| \leq \frac{1}{2}$. As previously, it is sufficient by antisymmetry (since $\mathrm{sinc}'(-x) = -\mathrm{sinc}'(x)$) to prove the result on $x \in \mathbb{R}_+$. We proceed by studying the function $\rho : x \in \mathbb{R}_+ \mapsto \frac{1}{2}x^2 - x\cos(x) + \sin(x)$, null at zero, whose derivative is $\rho'(x) = x + x\sin(x) - \cos(x) + \cos(x) = x(1 + \sin(x)) \geq 0$. Hence $\rho(x) \geq \rho(0) = 0$ and thus $\frac{1}{2}x^2 \geq x\cos(x) - \sin(x)$, therefore $\frac{1}{2} \geq \mathrm{sinc}'(x)$ for $x \in \mathbb{R}_+$. Similarly for the other inequality, study $\tau : x \in \mathbb{R}_+ \mapsto -\frac{1}{2}x^2 - x\cos(x) + \sin(x)$, whose derivative is $\tau'(x) = -x(1 - \sin(x)) \leq 0$, hence $\tau(x) \leq \tau(0) = 0$, thus $-\frac{1}{2}x^2 \leq x\cos(x) - \sin(x)$, therefore $\mathrm{sinc}'(x) \geq -\frac{1}{2}$ for $x \in \mathbb{R}_+$. This concludes the proof that $x \geq 0 \Rightarrow |\mathrm{sinc}'(x)| \leq \frac{1}{2}$.

Computing the derivative again,

$$\mathrm{sinc}'' : x \mapsto \frac{2\sin(x)}{x^3} - \frac{2\cos(x)}{x^2} - \frac{\sin(x)}{x}$$

Using the recently proven fact $|\mathrm{sinc}'(x)| \leq 1/2$,

$$\left|\mathrm{sinc}''(x)\right| = \left|-2\frac{\mathrm{sinc}'(x)}{x} - \frac{\sin(x)}{x}\right| \leq 2\left|\frac{\mathrm{sinc}'(x)}{x}\right| + \left|\frac{\sin(x)}{x}\right| \leq \frac{2}{|x|}$$

It remains to show that $\psi$ is increasing on $[0, x_0]$ and $\phi$ is decreasing on the same interval. Since $\phi$ is continuous, $\phi(0) = \frac{1}{3}$ and $x_0$ is the first zero of $\phi$ by definition, it follows that $\psi'(x) = \phi(x) \geq 0$ for $x \in [0, x_0]$, which proves the first statement.

It remains to show that $\phi = -\mathrm{sinc}''$ is decreasing on $[0, x_0]$, for which it is sufficient to show that $\mathrm{sinc}'''$ is positive on this interval. Using the form $\mathrm{sinc}''(x) = -2\mathrm{sinc}'(x)/x - \mathrm{sinc}(x)$,

$$\begin{aligned}
\mathrm{sinc}'''(x) &= -2\frac{\mathrm{sinc}''(x)}{x} + 2\frac{\mathrm{sinc}'(x)}{x^2} - \mathrm{sinc}'(x) \\
&= -2\frac{1}{x}\left(-2\frac{\mathrm{sinc}'(x)}{x} - \mathrm{sinc}(x)\right) + 2\frac{\mathrm{sinc}'(x)}{x^2} - \mathrm{sinc}'(x) \\
&= \left(\frac{6}{x^2} - 1\right)\mathrm{sinc}'(x) + 2\frac{\mathrm{sinc}(x)}{x}
\end{aligned}$$

on the interval $[0, x_0]$, $\mathrm{sinc}(x) \geq 0$, $\mathrm{sinc}'(x) \geq 0$ and $(6/x^2 - 1) \geq 0$.

## A.7 Kurdyka-Łojasiewicz region for two-layer networks

We split the proof of Proposition 4.6 in two parts, first the inequality satisfied in high probability (a), and then how to leverage this inequality to get a convergence speed (b).

*Proof of Proposition 4.6 (a).* For the first part of the proof (the inequality), note that by density of $\mathrm{Im}(F)$ in $L^1(K)$ [Leshno et al., 1993, Proposition 1], there exists $m^* \in \mathbb{N} \setminus \{0\}$ and $\theta^* \in \Theta^{(m^*)}$ such that $\sup_{x \in K} \|F(\theta^*)(x) - f^*(x)\| \leq \sqrt{\varepsilon}/2$. We will write $\|g\|_\infty = \sup_{x \in K} |g(x)|$ for shortness.

Let $(w^*, a^*) = \theta^*$. We will show that for any $(w, a)$ such that there is at least one $w_i$ in a bassin around $w_j^*$ for all $j \in [m^*]$, $F$ has a first-order approximation that is an $\varepsilon$-approximation of $f^*$ (i.e. it is sufficient to roughly approximate features to get relatively good gradients far from the optimum).

Formally, let $\eta = \sqrt{\varepsilon}/(2\|a^*\|_1 L_\sigma D)$, where $L_\sigma$ is the Lipschitz constant of $\sigma$ and $D = \sup_{x \in K}\|x\|_2$. Let $\mathcal{P}_{\theta^*}^{(m)} = \{(w, a) \in \Theta^{(m)} \mid \forall i \in [m^*], \exists j \in [m], \|w_j - w_i^*\|_2 \leq \eta\}$.

Let us show that if $\theta \in \mathcal{P}_{\theta^*}^{(m)}$, then $\exists \nu \in \Theta^{(m)}$ such that $\|F(\theta) + \mathrm{d}F_\theta \cdot \nu - f^*\|_\infty \leq \sqrt{\varepsilon}$.

Let $m \in \mathbb{N}$ and $(w, a) \in \mathcal{P}_{\theta^*}^{(m)}$. For all $i \in [m^*]$, define $j_i \in \mathrm{argmin}_{k \in [m]}\|w_i^* - w_k\|_2$. In words, $j_i \in [m]$ is the index of the (learned) neuron closest to target neuron $i \in [m^*]$. Then, let $\nu_0 = (-a_k + \sum_{i \in [m^*]} \delta_{k=j_i} a_i^*, 0)_{k \in [m]} \in \Theta^{(m)}$. Observe that for $x \in \mathbb{R}^d$,

$$(F(w, a) + \mathrm{d}F_{(w,a)} \cdot \nu_0)(x) = \sum_{k \in [m]} a_k \sigma(w_k \cdot x) + \sum_{k \in [m]} \left(-a_k + \sum_{i \in [m^*]} \delta_{k=j_i} a_i^*\right) \sigma(w_k \cdot x)$$

$$= \sum_{i \in [m^*]} a_i^* \, \sigma(w_{j_i} \cdot x)$$

Therefore, using the Lipschitz property of $\sigma$, then $\|w_{j_i} - w_i^*\|_2 \leq \eta$,

$$\|F(w, a) + \mathrm{d}F_{(w,a)} \cdot \nu_0 - f^*\|_\infty \leq \|F(w, a) + \mathrm{d}F_{(w,a)} \cdot \nu_0 - F(w^*, a^*)\|_\infty + \|F(w^*, a^*) - f^*\|_\infty$$

$$\leq \sup_{x \in K}\left|\sum_{i \in [m^*]} a_i^* \, (\sigma(w_{j_i} \cdot x) - \sigma(w_i^* \cdot x))\right| + \frac{\sqrt{\varepsilon}}{2}$$

$$\leq \sum_{i \in [m^*]} |a_i^*| \, L_\sigma \|w_{j_i} - w_i^*\|_2 \sup_{x \in K}\|x\|_2 + \frac{\sqrt{\varepsilon}}{2}$$

$$\leq \|a^*\|_1 L_\sigma \frac{\sqrt{\varepsilon}}{2\|a^*\|_1 L_\sigma D} D + \frac{\sqrt{\varepsilon}}{2}$$

$$\leq \sqrt{\varepsilon}$$

Moreover, observe that $\|\nu_0\|_2 \leq \|-a\|_2 + \|a^*\|_2 = \|a\|_2 + \|a^*\|_2$.

Then, similarly to the linear cases, define the functional quadratic loss $\ell : f \mapsto \|f - f^*\|_{\mathcal{D}}^2$, which satisfies the Polyak-Łojasiewicz inequality $\|\nabla \ell_f\|_{\mathcal{D}}^2 \geq 4\ell(f)$. It remains to transfer it to $\mathcal{L}$. Unfortunately, we will not be able to lower-bound the Rayleigh quotient by a constant, so we perform a slightly different manipulation to obtain a Kurdyka-Łojasiewicz inequality on $\mathcal{L}$ anyway.

$$\|\nabla \mathcal{L}(w, a)\|_2^2 = \mathrm{R}\left(K_\theta^\star; \nabla \ell_{F(w,a)}, \nabla \ell_{F(w,a)}\right) \|\nabla \ell_{F(w,a)}\|_{\mathcal{D}}^2$$

$$= \sup_{\nu \in \Theta^{(m)} \setminus \{0\}} \mathrm{R}\left(\mathrm{d}F_{(w,a)}^\star; \nu, \nabla \ell_{F(w,a)}\right)^2 \|\nabla \ell_{F(w,a)}\|_{\mathcal{D}}^2$$

$$= \sup_{\nu \in \Theta^{(m)} \setminus \{0\}} \frac{\langle \mathrm{d}F_\theta \cdot \nu, \nabla \ell_f\rangle_{\mathcal{D}}^2}{\|\nu\|_2^2}$$

$$\geq \frac{\langle \mathrm{d}F_\theta \cdot \nu_0, 2(F(\theta) - f^*)\rangle_{\mathcal{D}}^2}{\|\nu_0\|_2^2}$$

These computations are similar to the other cases, but since we're unable to obtain a lower bound multiplicatively by bounding the Rayleigh quotient directly, we instead split it to accept an additive $\varepsilon$ term (leading to convergence to $\varepsilon$ instead of convergence to zero as in the other examples).

$$\|\nabla\mathcal{L}(w,a)\|_2^2 \geq \frac{1}{\|\nu_0\|_2^2}\left(\|F(\theta)+\mathrm{d}F_\theta\cdot\nu_0-f^*\|_{\mathcal{D}}^2 - \|\mathrm{d}F_\theta\cdot\nu_0\|_{\mathcal{D}}^2 - \|F(\theta)-f^*\|_{\mathcal{D}}^2\right)^2 \quad (1)$$

$$= \frac{1}{\|\nu_0\|_2^2}\left(\mathcal{L}(\theta)+\|\mathrm{d}F_\theta\cdot\nu_0\|_{\mathcal{D}}^2 - \|F(\theta)+\mathrm{d}F_\theta\cdot\nu_0-f^*\|_{\mathcal{D}}^2\right)^2$$

$$\geq \frac{1}{\|\nu_0\|_2^2}\left(\mathcal{L}(\theta)+0-\varepsilon\right)_+^2 \quad (2)$$

$$\geq \frac{1}{(\|a\|_2+\|a^*\|_2)^2}\left(\mathcal{L}(\theta)-\varepsilon\right)_+^2$$

Where (1) is the parallelogram identity for the $\ell_2$ norm, $2\langle u,v\rangle = \|u+v\|_2^2 - \|u\|_2^2 - \|v\|_2^2$, and where (2) is $(u \geq v \geq 0) \Rightarrow (u^2 \geq v^2)$ when $\mathcal{L}(\theta) \geq \varepsilon$. This almost concludes the first part of the proof (the inequality), though it remains to show that $(\|a\|_2+\|a^*\|_2)$ is bounded by $(\|a-a_0\|_2+C)$ for some constant $C \in \mathbb{R}_+^*$ independent of $m$, with high probability. For reasons that will become apparent later, let $\delta_0 = \delta/2 \in\,]0,1[$. We now focus on the high probability part of the proof.

We have proved so far that under some condition on $\theta$, $\mathcal{L}$ satisfies a Kurdyka-Łojasiewicz inequality at $\theta$. It remains to prove that this condition is satisfied with high probability near initialization. For any $\theta_0 \in \Theta^{(m)}$, let $\mathcal{B}(\theta_0, R) = \{\theta \in \Theta^{(m)} \mid \|\theta - \theta_0\|_2 \leq R\}$ be the $R$-radius ball around $\theta_0$. We would like to show that for some $m \in \mathbb{N}$, it holds

$$\mathbb{P}_{\theta_0\sim\mathcal{I}_m}\left(\mathcal{B}(\theta_0,R)\subseteq\mathcal{P}_{\theta^*}^{(m)}\right)\geq 1-\delta_0$$

To prove this statement, we will need a stronger property than just $\theta_0 \in \mathcal{P}_{\theta^*}^{(m)}$ with high probability. Namely, let $\mathcal{Q}_{\theta^*}^{(m)} = \{\theta \in \Theta^{(m)} \mid \forall i \in [m^*], |\{j \in [m], \|w_j - w_i^*\|_2 \leq \frac{\eta}{2}\}| \geq k\}$ be the set of parameters such that there are at least $k$ neurons in each (half smaller) feature bassin, for some yet unspecified value of $k \in \mathbb{N}^*$. In the set $\mathcal{P}_{\theta^*}$ we only required that $k=1$ and allowed larger bassins.

Let $(H_u \subseteq [m])_{u\in[k]}$ be any partition of $[m]$ into $k$ sets, each of size at least $\lfloor m/k \rfloor$. For (2) hereafter, note that if a set $S \subseteq [m]$ has size $|S| < k$, then $\exists u \in [k], S \cap H_u = \emptyset$, by the pigeonhole principle.

$$\mathbb{P}_{\theta_0\sim\mathcal{I}_m}\left(\theta_0\notin\mathcal{Q}_{\theta^*}^{(m)}\right) = \mathbb{P}_{(w,a)\sim\mathcal{I}_m}\left(\exists i\in[m^*], \left|\left\{j\in[m], \|w_j-w_i^*\|_2\leq\frac{\eta}{2}\right\}\right| < k\right)$$

$$\leq \sum_{i=1}^{m^*}\mathbb{P}\left(\left|\left\{j, \|w_j-w_i^*\|_2\leq\frac{\eta}{2}\right\}\right| < k\right) \quad (1)$$

$$\leq \sum_{i=1}^{m^*}\mathbb{P}\left(\exists u\in[k], \forall j\in H_u, \|w_j-w_i^*\|_2>\frac{\eta}{2}\right) \quad (2)$$

$$\leq \sum_{i=1}^{m^*}\sum_{u\in[k]}\prod_{j\in H_u}\mathbb{P}\left(\|w_j-w_i^*\|_2>\frac{\eta}{2}\right) \quad (3)$$

$$\leq \sum_{i=1}^{m^*}k\left(\mathbb{P}_{y\sim\mathcal{N}(0_d,1_d)}\left(\|y-w_i^*\|_2>\frac{\eta}{2}\right)\right)^{\lfloor m/k\rfloor}$$

Where (1) and (3) are union bounds, followed by independent identical distribution of $w_j$.

For all $i \in [m^*]$, it holds $\mathbb{P}_{y\sim\mathcal{N}(0_d,1_d)}\left(\|y-w_i^*\| > \eta/2\right) < 1$ (i.e. full support), therefore for any fixed constant $k$, there exists an $m$ sufficiently large such that it holds $\mathbb{P}_{\theta_0\sim\mathcal{I}_m}(\theta_0\notin\mathcal{Q}_{\theta^*}^{(m)})\leq\delta_0$.

Let $\theta_0 \in \mathcal{Q}_{\theta^*}^{(m)}$. Let us show that $\mathcal{B}(\theta_0, R) \subseteq \mathcal{P}_{\theta^*}^{(m)}$. Let $\theta \in \mathcal{B}(\theta_0, R)$, and $i \in [m^*]$. We write $(w^{(0)}, a^{(0)}) = \theta_0$ the two components of $\theta_0$. By assumption, there is a subset $J \subseteq [m]$ of size $|J| = k$ such that $\forall j \in J, \|w_j^{(0)} - w_i^*\|_2 \leq \frac{\eta}{2}$.

$$\min_{j \in [m]} \|w_j - w_i^*\|_2 \le \min_{j \in J} \|w_j - w_i^*\|_2 \le \frac{1}{k} \sum_{j \in J} \|w_j - w_i^*\|_2$$

$$\le \frac{1}{k} \sum_{j \in J} \|w_j^{(0)} - w_i^*\|_2 + \|w_j - w_j^{(0)}\|_2$$

$$\le \frac{\eta}{2} + \frac{1}{k} \sum_{j \in J} \|w_j - w_j^{(0)}\|_2$$

$$\le \frac{\eta}{2} + \sqrt{\frac{1}{k} \sum_{j \in J} \|w_j - w_j^{(0)}\|_2^2}$$

$$\le \frac{\eta}{2} + \frac{R}{\sqrt{k}}$$

Thus if $\sqrt{k} \ge 2R/\eta$, it holds that $\forall m$, $\theta_0 \in \mathcal{Q}_{\theta^*}^{(m)} \Rightarrow \mathcal{B}(\theta_0, R) \subseteq \mathcal{P}_{\theta^*}^{(m)}$. In particular, there exists $m$ such that $\mathbb{P}_{\theta_0 \sim \mathcal{I}_m}\left(\theta_0 \in \mathcal{Q}_{\theta^*}^{(m)}\right) \ge 1 - \delta_0$, thus $\mathbb{P}_{\theta_0 \sim \mathcal{I}_m}\left(\mathcal{B}(\theta_0, R) \subseteq \mathcal{P}_{\theta^*}^{(m)}\right) \ge 1 - \delta_0$, as claimed.

As previously noted, it remains to show that $(\|a\|_2 + \|a^*\|_2) \le \|a - a^{(0)}\|_2 + C$, for a constant $C$ independent of $m$. Let $C = \sqrt{1/\delta_0} + \|a^*\|_2$. The norm $\|a^*\|_2$ depends on $\varepsilon$, and thus the "optimal" number of neurons $m^*$, but not on the number of "training" neurons $m$. To reach the conclusion, let us show that $\mathbb{P}_{\theta_0 \sim \mathcal{I}_m}\left(\sup_{(w,a) \in \mathcal{B}(\theta_0, R)} \|a\|_2 - \|a - a^{(0)}\|_2 \le \sqrt{1/\delta_0}\right) \ge 1 - \delta_0$.

$$\mathbb{P}_{\theta_0 \sim \mathcal{I}_m}\left(\sup_{(w,a) \in \mathcal{B}(\theta_0, R)} \|a\|_2 - \|a - a^{(0)}\| \ge \sqrt{\frac{1}{\delta_0}}\right)$$

$$\le \mathbb{P}_{a^{(0)} \sim \mathcal{N}\left(0_m, I_m/\sqrt{m}\right)}\left(\sup_{a \in \mathcal{B}(a^{(0)}, R)} \|a\|_2 - \|a - a^{(0)}\|_2 \ge \sqrt{\frac{1}{\delta_0}}\right)$$

$$\le \mathbb{P}_{a^{(0)} \sim \mathcal{N}\left(0_m, I_m/\sqrt{m}\right)}\left(\|a^{(0)}\|_2 \ge \sqrt{\frac{1}{\delta_0}}\right) \tag{1}$$

$$= \mathbb{P}_{a^{(0)} \sim \mathcal{N}\left(0_m, I_m/\sqrt{m}\right)}\left(\sum_{i \in [m]} \left(a_i^{(0)}\right)^2 \ge \frac{1}{\delta_0}\right)$$

$$\le \frac{\mathbb{E}_{a^{(0)}}\left[\sum_{i \in [m]} \left(a_i^{(0)}\right)^2\right]}{1/\delta_0} = \delta_0 \sum_{i \in [m]} \frac{1}{m} = \delta_0 \tag{2}$$

Where (1) is because $\|a^{(0)}\|_2 \ge \|a\|_2 - \|a - a^{(0)}\|_2$ by triangular inequality, therefore for all constants $M \in \mathbb{R}_+^*$, it holds $\{\|a\|_2 - \|a - a^{(0)}\|_2 \ge M\} \subseteq \{\|a^{(0)}\|_2 \ge M\}$, and where (2) is Markov's inequality. Now, tying all pieces together,

$$\mathbb{P}_{\theta_0 \sim \mathcal{I}_m}\left(\forall \theta \in \mathcal{B}(\theta_0, R), \|\nabla \mathcal{L}(\theta)\|_2^2 \ge \frac{1}{(\|\theta - \theta_0\|_2 + C)^2}\left(\mathcal{L}(\theta) - \varepsilon\right)_+^2\right)$$

$$\ge \mathbb{P}_{\theta_0 \sim \mathcal{I}_m}\left(\left(\theta_0 \in \mathcal{Q}_{\theta^*}^{(m)}\right) \cap \left(\sup_{(w,a) \in \mathcal{B}(\theta_0, R)} \|a\|_2 - \|a - a^{(0)}\|_2 \le \sqrt{\frac{1}{\delta_0}}\right)\right)$$

$$\ge 1 - \mathbb{P}_{\theta_0 \sim \mathcal{I}_m}\left(\theta_0 \notin \mathcal{Q}_{\theta^*}^{(m)}\right) - \mathbb{P}_{(w^{(0)}, a^{(0)}) \sim \mathcal{I}_m}\left(\sup_{a \in \mathcal{B}(a^{(0)}, R)} \|a\|_2 - \|a - a^{(0)}\|_2 > \sqrt{\frac{1}{\delta_0}}\right)$$

$$\ge 1 - \delta_0 - \delta_0 = 1 - \delta$$

This completes the proof that the Kurdyka-Łojasiewicz inequality holds on a ball near the initialization with high probability over the initialization when the number of neurons is sufficiently large. $\qquad\square$

The idea for the second part of the proof is to put the Kurdyka-Łojasiewicz inequality in separable form, then integrate it (following Scaman et al. [2022, Proposition 4.6], but we will reproduce the proof for shortness). This will yield one upper bound on the loss if the weights remain in the ball, and an other bound on the loss if the weights escape the ball, which we can force into coinciding with the desired precision by adjusting the chosen radius $R$.

*Proof of Proposition 4.6 (b).* Part (a) of this proof has established the following proposition w.h.p:

$$\exists c \in \mathbb{R}_+, \forall R \in \mathbb{R}_+, \exists m \in \mathbb{N}^*, \forall \theta \in \mathcal{B}(\theta_0, R), \quad \|\nabla \mathcal{L}(\theta)\|_\Theta^2 \geq \frac{(\mathcal{L}(\theta) - \varepsilon)_+^2}{(\|\theta - \theta_0\|_2 + c)^2}$$

Where the probability is taken over initializations $\theta_0 \sim \mathcal{I}_m$. Moreover, with high probability as well, $\mathcal{L}(\theta_0) \leq L_0 \in \mathbb{R}_+$ (independently of $m \in \mathbb{N}$, see Lemma A.6 below for details).

Let $\varepsilon_0 \in \mathbb{R}_+^*$ be any target precision. Let $R > 2cL_0/\varepsilon_0$ and apply Proposition 4.6 (a) with $\varepsilon = \frac{\varepsilon_0}{2}$.

Let $\theta : \mathbb{R}_+ \to \Theta$ be a gradient flow of $\mathcal{L}$ with $\theta(0) = \theta_0$. Since $t \mapsto \mathcal{L}(\theta_t)$ is a non-negative non-increasing function of time, it must converge to a non-negative real value $\mathcal{L}(\theta_t) \to_t \eta \in \mathbb{R}_+$ (by monotone convergence). Therefore, let us show that it will reach a loss below $\varepsilon_0$, which is sufficient to obtain $\eta \leq \varepsilon_0$. If $\mathcal{L}(\theta_0) \leq \varepsilon_0$ then the proof is concluded, otherwise let us define $T = \inf\left(\{t \in \mathbb{R}_+ \mid \theta_t \in \mathcal{B}(\theta_0, R)\} \cap \{t \in \mathbb{R}_+ \mid \mathcal{L}(\theta_t) \geq \varepsilon_0\}\right) \in \mathbb{R}_+^* \cup \{+\infty\}$. We will now focus our attention to the interval $I = [0, T[$, where the Kurdyka-Łojasiewicz inequality is satisfied (by definition of $T$). We start by weakening the inequality to get rid of $\|\theta - \theta_0\|_2$ by separability.

Define $r : [0, T[ \to \mathbb{R}_+$, as $r : t \mapsto \int_0^t \|\partial_t \theta(u)\| \, du$. Observe that for all $t < T$, it holds $\|\theta_t - \theta_0\| \leq r_t$ by triangular inequality. Additionally, using the square root of the Kurdyka-Łojasiewicz inequality,

$$\partial_t r_t = \|\partial_t \theta\|_2 = \|\nabla \mathcal{L}(\theta_t)\|_2 = \frac{\|\nabla \mathcal{L}(\theta_t)\|_2^2}{\|\nabla \mathcal{L}(\theta_t)\|_2} \leq \frac{\|\nabla \mathcal{L}(\theta_t)\|_2^2}{\frac{1}{r_t + c}(\mathcal{L}(\theta_t) - \varepsilon)} = (r_t + c)\frac{-\partial_t \mathcal{L}(\theta)}{\mathcal{L}(\theta_t) - \varepsilon}$$

This corresponds to the inequality $\partial_t(\psi \circ r) \leq \partial_t(\varphi \circ \mathcal{L})$, with desingularizers $\varphi : u \mapsto -\log(u - \varepsilon)$ and $\psi : u \mapsto \log(u + c)$. Integrating between $0$ and $t < T$, this yields the inequality

$$\log\left(\frac{r_t + c}{c}\right) = \left[\log(r_u + c)\right]_0^t \leq \left[-\log(\mathcal{L}(\theta_u) - \varepsilon)\right]_0^t = \log\left(\frac{\mathcal{L}(\theta_0) - \varepsilon}{\mathcal{L}(\theta_t) - \varepsilon}\right)$$

$$r_t + c \leq c \frac{\mathcal{L}(\theta_0) - \varepsilon}{\mathcal{L}(\theta_t) - \varepsilon} \tag{1}$$

Define the $\varepsilon$-discounted loss $\mathcal{L}^\varepsilon : \Theta \to \mathbb{R}_+$ as $\mathcal{L}^\varepsilon : u \mapsto (\mathcal{L}(u) - \varepsilon)_+$. For all $t < T$, it holds $\mathcal{L}(\theta_t) = \mathcal{L}^\varepsilon(\theta_t) + \varepsilon$, thus $\nabla \mathcal{L}(\theta_t) = \nabla \mathcal{L}^\varepsilon(\theta_t)$. Therefore the restriction $\theta_I : [0, T[ \to \Theta$ is a gradient flow of $\mathcal{L}^\varepsilon$. Moreover, injecting inequality (1) above into the previous Kurdyka-Łojasiewicz inequality, we get the more easily understood inequality

$$\forall t < T, \quad \|\nabla \mathcal{L}^\varepsilon(\theta_t)\|_2^2 \geq \frac{1}{(c\,\mathcal{L}^\varepsilon(\theta_0))^2}(\mathcal{L}^\varepsilon(\theta_t))^4$$

Setting $\kappa = 3/(c\,\mathcal{L}^\varepsilon(\theta_0))^2 \in \mathbb{R}_+^*$, and the desingularizer $\varphi : u \mapsto -1/u^3$, this corresponds to the inequality $d\varphi_{\mathcal{L}^\varepsilon}\|\nabla \mathcal{L}^\varepsilon\|_2^2 \geq \kappa$. Integrating between $0$ and $T$ according to Proposition 3.1, this gives at all times $t < T$, the inequality $\mathcal{L}^\varepsilon(\theta_t) \leq (\mathcal{L}^\varepsilon(\theta_0)^{-3} + \kappa t)^{-1/3}$. We are now ready to conclude by case disjunction. If $T = +\infty$ then it is immediate that the convergence speed holds for $t \in \mathbb{R}_+$. If $T < +\infty$, there are two cases to tackle. If $\mathcal{L}(\theta_T) \leq \varepsilon_0$, then since the loss is decreasing, it holds for all $t \geq T$ that $\mathcal{L}(\theta_t) \leq \mathcal{L}(\theta_T) \leq \varepsilon_0$, therefore the bound holds for $t \geq T$ as well which concludes this case. If $\|\theta_T - \theta_0\|_2 = R$, then equation (1) gives $\mathcal{L}(\theta_T) - \varepsilon \leq \mathcal{L}(\theta_0) c/(R + c) < \varepsilon$ (by definition of $R$), thus $\mathcal{L}(\theta_T) \leq \varepsilon + \varepsilon = \varepsilon_0$, and therefore by the same argument, it holds for $t \geq T$ that $\mathcal{L}(\theta_t) \leq \mathcal{L}(\theta_T) \leq \varepsilon_0$ and thus the bound is extended to $t \in \mathbb{R}_+$, which concludes the proof. $\quad\square$

**Lemma A.6** (Bounded initial loss with high probability). *Under the hypotheses of Proposition 4.6, for all $\delta \in ]0, 1[$, there exists $L_0 \in \mathbb{R}_+$, such that for all $m \in \mathbb{N}^*$, it holds $\mathbb{P}_{\theta_0} \left( \mathcal{L}(\theta_0) \leq L_0 \right) \geq 1 - \delta$.*

*Proof.* For an $m \in \mathbb{N}^*$, let $a_i \sim \mathcal{N}(0, 1/\sqrt{m})$ for $i \in [m]$ and $w_{i,j} \sim \mathcal{N}(0, 1)$ for $(i, j) \in [m] \times [d]$ be independent random variables, so that $(a, w) \sim \mathcal{I}_m$. We wish to prove that $(\mathcal{L}(a, w) \leq L_m)$ holds with high probability for some constant $L_m \in \mathbb{R}_+$. First, observe that

$$\mathcal{L}(a, w) = \mathbb{E}_{x \sim \mathcal{D}} \left[ \left( \sum_{i \in [m]} a_i \sigma(w_i \cdot x) - f^*(x) \right)^2 \right] = \|f_\theta - f^*\|_{\mathcal{D}}^2 \leq 2 \left( \|f_\theta\|_{\mathcal{D}}^2 + \|f^*\|_{\mathcal{D}}^2 \right)$$

Since $\|f^*\|_{\mathcal{D}}^2$ is a constant, it is sufficient to show that $\|f_\theta\|_{\mathcal{D}}^2$ is bounded with high probability.

In order to proceed by Markov's inequality, let us show that $\mathbb{E}_{a,w} \left[ \mathbb{E}_x \left[ \left( \sum_i a_i \sigma(w_i \cdot x) \right)^2 \right] \right]$ is finite. First, if $i \neq j$, then $\mathbb{E}_{a,w,x} \left[ a_i a_j \sigma(w_i x) \sigma(w_j x) \right] = \mathbb{E}[a_i] \mathbb{E}[a_j] \mathbb{E} \left[ \sigma(w_i \cdot x) \sigma(w_j \cdot x) \right] = 0$, by independence of $a$ an $(w, x)$, and independence of $a_i$ and $a_j$. Therefore

$$\mathbb{E}_{a,w,x} \left[ \left( \sum_{i \in [m]} a_i \sigma(w_i \cdot x) \right)^2 \right] = \sum_{i \in [m]} \mathbb{E}_{a,w,x} \left[ a_i^2 \sigma(w_i \cdot x)^2 \right]$$

$$= \sum_{i \in [m]} \mathbb{E}_a \left[ a_i^2 \right] \mathbb{E}_{w,x} \left[ \sigma(w_i \cdot x)^2 \right] \tag{1}$$

$$\leq \sum_{i \in [m]} \frac{1}{m} \mathbb{E}_{w,x} \left[ \left( |\sigma(0)| + L_\sigma |w_i \cdot x| \right)^2 \right] \tag{2}$$

$$\leq \frac{1}{m} \sum_{i \in [m]} \mathbb{E}_{w,x} \left[ \left( |\sigma(0)| + L_\sigma \|w_i\|_2 \|x\|_2 \right)^2 \right] \tag{3}$$

$$\leq \frac{1}{m} \sum_{i \in [m]} \mathbb{E}_{w,x} \left[ \left( |\sigma(0)| + L_\sigma \|w_i\|_2 D \right)^2 \right] \tag{4}$$

$$\leq \frac{1}{m} \sum_{i \in [m]} 2 \left( \sigma(0)^2 + L_\sigma^2 D^2 \mathbb{E}_w \left[ \|w_i\|_2^2 \right] \right) \tag{5}$$

$$= \frac{1}{m} \sum_{i \in [m]} 2 \left( \sigma(0)^2 + L_\sigma^2 D^2 \mathbb{E}_w \left[ \sum_j w_{i,j}^2 \right] \right)$$

$$= \frac{1}{m} \sum_{i \in [m]} 2 \left( \sigma(0)^2 + L_\sigma^2 D^2 d \right) = 2 \left( \sigma(0)^2 + L_\sigma^2 D^2 d \right)$$

Where (1) is independence, (2) is because $\sigma$ is $L_\sigma$-Lipschitz, (3) is Cauchy-Schwarz, (4) is bounded input radius $D = \sup_{x \in K} \|x\|_2$ by compact-support assumption, (5) is $(u + v)^2 \leq 2(u^2 + v^2)$, and the remaining is evaluation in closed form.

Let $K = 2 \left( \sigma(0)^2 + L_\sigma^2 D^2 d \right) / \delta \in \mathbb{R}_+^*$. By Markov's inequality, $\mathbb{P}_\theta \left( \|f_\theta\|_{\mathcal{D}}^2 \geq K \right) \leq \mathbb{E}_\theta \left[ \|f_\theta\|_{\mathcal{D}}^2 \right] / K = \delta$. This constant does not depend on $m$, therefore the choice of bound $L_0 = 2 \left( K + \|f^*\|_{\mathcal{D}}^2 \right)$ concludes the proof. $\square$