# OpenReview forum: "Convergence beyond the over-parameterized regime using Rayleigh quotients"
_NeurIPS.cc/2022/Conference — NeurIPS 2022 Accept_

### Official Review · Reviewer_xNfg · 2022-07-07

**Rating:** 7
**Confidence:** 4
**Soundness:** 3 good
**Presentation:** 3 good
**Contribution:** 3 good

**Summary:**

This work investigates the convergence of gradient flow to zero training (and test) loss. The main idea is to derive Kurdyka-Lojasiewicz inequality which is accomplished by lower bounding Rayleigh quotients, and several showcases of common learning problems have been analyzed by this technique.

**Questions:**

Typo:
line 319, "be compact". line 703, "has a ...". line 730, equation (1), it should be "l.h.s $\leq \frac{\eta}{2}$". in equation just before line 737, it might be $\frac{\eta}{2} + \frac{R}{\sqrt{k}}$, which yields $\sqrt{k}\geq 2R/\eta$. line 360, "there there". line 545, there might be an error in calculating first component of the gradient.

Questions and suggestions:

1- Proof and statement of Prop 4.6 needs revision. First I can't find a proof that $\theta(\mathbb{R}_+)\subseteq \mathcal{B}(\theta_0,R)$ if $\theta$ is a gradient flow, and in fact you just assume this condition holds in line 745 without proving it and further in line 328 you defer its proof to future work. Please indicate what is the difficulty in proving it. Most importantly, in the beginning of page 27, the conclusion in the 3rd line $\cdots \geq \frac{1}{\lVert \nu_0 \rVert^2}(\mathcal{L}(\theta_0) +0 - \epsilon)^2 $ does not seem correct because $a\geq b $ does not imply $a \geq b$. However it seems that there is a fix for that because you just need to consider $\lVert \nabla \mathcal{L}(w,a) \rVert_2 \geq \cdots \geq \frac{1}{\lVert \nu_0 \rVert}(\mathcal{L}-\epsilon)$. Even with this change, the argument in line 751 needs explanation on how you use prop 3.1 to show $\overset{\sim}{\mathcal{L}}(\theta)$ converges to zero. One way I can think of is to use $\phi(u)=-\frac{1}{u}$ which is a strictly increasing differentiable function on positive real numbers and $\phi'(u)=\frac{1}{u^2}$ and it gives $\lVert \nabla \tilde{\mathcal{L}}(\theta) \rVert^2 \geq \mu \tilde{\mathcal{L}}(\theta)^2$ which is true here as you assume $\mathcal{L}(\theta_t)\geq \epsilon$. Hence by showing that $\mathcal{L}(\theta_t)$ converges to $\eta\leq \epsilon$, it follows that $\mathcal{L}(\theta_t) - \epsilon$ is negative for large enough t, and hence $a \geq b$ does not imply $a^2 \geq b^2$ with $a=\lVert \nabla \mathcal{L}(\theta) \rVert$ and $b=\kappa (\mathcal{L}(\theta) - \epsilon)$. Finally I would like to know about rate of convergence by using Prop 3.1. It seems to me Prop 3.1 can only be used to show contradiction, i.e. $\eta\leq\epsilon$, and I'm wondering if $\lVert \nabla \mathcal{L}(\theta) \rVert \geq \kappa (\mathcal{L}(\theta) - \epsilon) $ can be used with prop 3.1 to give any rate of convergence.

2- (when) is the $\kappa$-isolated condition in line 307 satisfied? I couldn’t find how it is proved anywhere in the Proof section, and stating under which conditions it is satisfied seems crucial. Also please discuss how fast is the convergence rate of prop 4.5.

3- It is true that existing a "non-zero" $\epsilon$-separating ray leads to unique label in softmax classifier and also interesting that separating ray with zero loss implies dirac labels. Please cite any existing work that considers this separating ray assumption, assuming you are not the first one.

4- in line 590, is the conclusion $\nabla \ell(u): x\rightarrow \nabla \ell_x(u(x))$ a result of Lemma A.1?

5- in line 583, please explain more thoroughly how "convergence of positive r.v.s in expectation to zero leads to some subsequence converging a.e. to zero".



**Ethics Review Area:**

["I don’t know"]

**Limitations:**

The limitations concerning conditions under which prop 4.6 holds is discussed in lines 328-331. However I couldn't find how $\kappa$-isolated in line 307 and $\epsilon$-separating ray in line 309 are satisfied. It is strongly suggested to discuss these in the appendix.

**Strengths And Weaknesses:**

Overall the paper is well-written and the main messages are easy to follow. However I had problems in understanding section 4.5 and its proofs, as well as prop 4.2, and couldn't check their correctness. Other results, except for prop 4.6 which might need slight modification discussed later, seem to be correct. The main weakness can be seen in discussing the conditions under which prop 4.6 and prop 4.5 hold, as addressed in below.

I like the argument that Prop 3.1 makes the loss evolution more tractable using the function $\phi$. Indeed in line 164 you provide $\phi$ when PL condition is satisfied and further in section 4.3 provide example for multi-class logistic regression. In Prop 3.3 you argue that functional-space KL inequality is easier to obtain and together with Rayleigh quotient bound they imply parametric-space KL inequality, hence able to use the convergence result of Prop 3.1. The techniques employed throughout Prop 3.4-3.6 to split the Rayleigh quotient bound seems powerful and its first simple use is shown in linear models with quadratic loss in Prop 4.1.

I would also like to know about any previous work that analyzes Cross-entropy minimization with linear models, assuming you are not the first paper to consider it.

Overall the references include major important works in related areas, and some of them deal with Gradient Descent and overparameterized models. Please pinpoint the works that do not have overparametrized assumptions and if possible, compare your work to them. For example in line 150, you mention that "this was introduced as an extension to PL inequality for linear convergence ...". Please include as many references as possible that revolve around using inequalities in Kurdyka [1998]. One suggestion is to defer the proof of prop 4.1 to appendix so that you have more space.

---

> ### Author Response · Authors · 2022-08-02
> **Response to Reviewer xNfg**
>
> Thank you for your time and (very) thorough reading of all proof details.
> We have updated our submission to include your corrections (pdf in updated supplementary material, changes in red).
>
> We have fixed all the typos and mistakes you found.
> A discussion of previous results on logistic regression has also been added to Section 4.3, and a reference with several examples of use of KL inequalities to Section 3.
>
> Regarding your first question on $\mathcal{L}(\theta) - \varepsilon$, the issue was that we operated under the (unstated) assumption that $\mathcal{L}(\theta) \geq \varepsilon$ corresponding to non-null loss in Proposition 3.1, which was a mistake on our part. For succinctness and readability, we have modified the inequality to use the positive part $\left(\mathcal{L}(\theta) - \varepsilon\right)_+$ which is null when $\mathcal{L}(\theta) \leq \varepsilon$, and thus fixes the issue with minimal changes.
>
> The property $\theta(\mathbb{R}_+) \subseteq \mathcal{B}(\theta_0, R)$ was not a claim but an assumption of our statement. In the final version, we have removed this assumption and strengthened our proposition, which should lift this concern as well. We have modified the proof to give a convergence rate as requested, and assume you will find the idea very unsurprising for it almost exactly matches your suggestion.
>
> We have added a discussion of $\kappa$-isolation and $\varepsilon$-separation in Appendix A.5.5, please let us know if it answers your questions.
> The property you singled out on L590 of the first submission is indeed not a direct consequence of the lemma, but is easily checked by the same proof technique.
> We have also added the missing reference on extraction of sequences converging almost surely. Convergence of positive variables in expectation to zero implies convergence in $L_1$, which in turn implies convergence in probability, from which almost sure convergence follows up to extraction.
>
> We are sorry for the readability issues in Sections 4.2 and 4.5.
> We hope the idea for the convergence on the lemniscate was clear despite your trouble with Prop 4.2, for we found this low-dimensional counterexample enlightening. The periodic signal recovery on the other hand is significantly harder to follow, but demonstrates that a good understanding of the problem (materialized here by knowledge of an approximate singular vector decomposition of the network map differential, leveraged with Prop 3.6) is key to reach proofs in the regime with near-optimal parameter count, which seems to be the most interesting. This could also serve as a bridge to signal processing results, to show that this line of work is not restricted to deep learning. Any suggested changes or general ideas you may have to improve the proposition readability or high-level exposition of these two sections is of course very welcome.

---

> > ### Comment · Reviewer_xNfg · 2022-08-06
> > **Comment 1**
> >
> > The comments/concerns/suggestions have been reflected in the rebuttal version. Also thanks for highlighting the updates for easier follow-up.
> >
> > Important notice:
> > You should double-check page limit but I don't think you can go over 9 page limit in rebuttal, and it may cause rejection!

---

### Official Review · Reviewer_a4Qu · 2022-07-09

**Rating:** 4
**Confidence:** 4
**Soundness:** 3 good
**Presentation:** 2 fair
**Contribution:** 2 fair

**Summary:**

This paper presents a strategy to prove the loss convergence of gradient flow for training loss of the form $\ell(F(\theta))$, where $F$ is the vector consisting of neural net outputs on each data point, and $\ell$ maps the outputs to the final loss value (e.g., $\ell$ can be the MSE to target values). Previously NTK-based analysis assumes that the smallest eigenvalue of the NTK matrix $K = DF \cdot DF^\top$ is bounded from below, then proves the loss convergence through establishing the PL condition under this assumption. In this paper, the authors avoid the assumption on the smallest eigenvalue and instead point out a proof strategy based on Rayleigh quotients of the NTK matrix and KL condition. This strategy is then applied to prove loss convergence bounds for linear regression, linear logistic regression, and some partial results are also given for two-layer net with squared loss.

**Questions:**

1. I wonder if the authors have some concrete examples showing that loss convergence bounds can indeed be deduced for nonlinear neural nets beyond the over-parameterized regime (without extra assumptions).
2. How does the loss convergence bound in the linear logistic regression case compare with prior works? How much is known about the loss convergence rate of linear logistic regression in the literature?
3. When does the isolation assumption hold in Proposition 4.5?

**Limitations:**

The major limitation has already been mentioned in Weakness 1. Besides that, this paper also has some limitations that are common in many optimization papers: (1) the dynamics of SGD and gradient flow can be very different; (2) the loss function can be non-smooth.

**Strengths And Weaknesses:**

### Strengths:

1. The proposed proof strategy does not require the smallest eigenvalue of the NTK matrix to be lower bounded, allowing analysis when the number of data is more than the number of parameters.
2. Applying the proposed proof strategy gives an alternative way to understand the loss convergence bound for linear regression.

### Weaknesses:

1. While it sounds more promising to establish loss convergence bounds beyond the over-parameterized regime, this paper does not prove any concrete theorem on nonlinear neural nets with fewer parameters than the number of data. In Sections 4.4 and 4.5, the authors give some partial results establishing loss convergence under the assumption that the gradient flow does not move far from the initial point. This assumption weakens the strength of the results because it seems to be a more interesting and important question that why gradient flow does not intend to move far away in search of a better solution. In other words, it is unclear how useful the proposed proof strategy is in terms of overcoming the various difficulties in analyzing loss convergence beyond over-parameterized regime.
2. The results on linear logistic regression are not well contextualized relative to prior works. The authors claim that the loss convergence bound in this case is new (Line 97), but it is unclear how it compares with previous works such as the $O(1/t)$ bound in Theorem 5 of [1] for binary classification (see also Appendix E for the multi-class case). These two bounds should be the same when $t \to \infty$, so the authors should compare them carefully in the non-asymptotic case.
3. The presentation of the current paper is not very understandable, since there are too many notations and the notations vary in different settings. I spent hours reading this paper to understand and memorize all these notations just to understand the basic idea. One main issue is that the definitions in Sections 2 and 3 include too many technical details that are unrelated to the idea of bounding Rayleigh quotients, e.g., the concept of D-seminorm is somewhat unnecessary because one can just replace the D-seminorm of $\nabla \ell_f$ with an expectation over the dataset, which is more accessible to the readers. To improve the presentation, I recommend the authors to either reduce the number of notations or add a warmup section that illustrates the proof strategy in a simple case (e.g., linear regression) before introducing the general setup.

[1]. https://www.jmlr.org/papers/volume19/18-188/18-188.pdf

---

> ### Author Response · Authors · 2022-08-02
> **Response to Reviewer a4Qu**
>
>
> Thank you for your time and dedication to this submission. We have updated it to address the concerns you raised (pdf in updated supplementary material, changes in red).
>
> We will discuss the major problem (Weakness 1) last.
> First, thank you very much for the references on the convergence speed of the logistic regression, we have updated Section 4.3 to properly credit these works, together with comments on the differences between these bounds and the result presented. We are not aware of any more precise results on the convergence of logistic regression than those newly inserted. Please let us know if you think of any other relevant results that we have also missed, we will further update this section if this is the case.
> The consistency with the $\mathcal{O}(1/t)$ asymptotic behavior is checked in Appendix A.5.4, the link between the (unquantified) separability assumption and our formulation is now discussed in Appendix A.5.5.
> We have also added an explicit comment on $(1/n)$-isolation for finite datasets of size $n$ after Proposition 4.5.
>
> Regarding the general readability of the paper, we have added a notation table in Appendix A.1, we hope this helps with the reading.
> We found further simplification of notations in early sections hard to incorporate. Since the extension to infinite data is not standard, it requires some care in the definitions to have well-formed statements.
> In particular, the use of the $\mathcal{D}$-seminorm, though hard to get used to, makes much more explicit the geometric intuition in the following, with functions treated as vectors with angles between them for instance.
> We believe this choice of notation is an opportunity to steer the discussion towards more geometrical arguments that will ultimately be easier to understand.
> Changes we could consider to lighten notations are inlining the definition of the $\mathcal{D}$-seminorm everywhere to remove that definition, at the risk of erasing the geometric intuition and lenghtening some statements, along with the removal of the primal definition of $K_\theta$ and discussion of the difference with the dual $K_\theta^\star$, to shorten that definition.
> If you feel that these (or other changes you may suggest) would significantly improve the readability of this submission, please let us know, but please also keep in mind that the page limit and the conciseness that it implies are a major constraint of ours.
> Any suggestions regarding specific simplifications or preferred choices of notation for the later sections are also welcome.
>
> Regarding the major limitation, identified as Weakness 1 in the review, we have strengthened Proposition 4.6 to lift the initialization-ball assumption. We can now prove (with high probability) convergence to an arbitrarily low loss value under no additional assumption. We believe this clears the concern regarding concrete new results on non-linear networks.
> It remains that this work is restricted to continuous-time noiseless differentiable optimization, as observed in the review, but we argue that this is a good first step towards a stronger theory.

---

### Official Review · Reviewer_csDk · 2022-07-10

**Rating:** 7
**Confidence:** 5
**Soundness:** 4 excellent
**Presentation:** 3 good
**Contribution:** 4 excellent

**Summary:**

This paper proposes a new insight in proving the convergence of the training of deep learning. Taking advantage of  the notion of Rayleigh quotients, the authors show that  KL inequalities for a broader set of neural network architectures and loss functions can be obtained. Moreover, several illustrative examples are provided to demonstrate their idea. In particular, their analysis does not require over-parameterization.

**Questions:**

Minor issues:

1 A table of notations is preferred.

2 What does the ``uniform conditioning" (in Line 19) mean?

**Limitations:**

YES

**Strengths And Weaknesses:**

My detailed comments are given as below.

Strengths:

1 This is a theoretical article on the convergence of the GD for deep learning. This topic has been intensively studied recently. Most of previous works require the over-parameterization. But this work is different, and it presents a new strategy in proving the convergence based on the Rayleigh quotients and the KL inequalities. This innovative idea has intrigued me, and I feel that this article contributes significantly to  the understanding of training dynamics of NNs.

2 One of main limitations of previous works is that their arguments require the positiveness of the NTK which further requires the over-parameterization. This paper points out that  one can bound the Rayleigh quotient of the gradient instead of lower bounding the eigenvalues of the NTK. Furthermore, the authors show the connection between the Rayleigh quotient and the KL inequalities which ensure the convergence.

3 This paper is very well written and easy to read. Several case studies are presented in Sec 4.These cases are common and representative, and make their ideas very convincing. Moreover, the proof seems clean. Although I did not thoroughly examine the proof, it provided me with several fresh perspectives.

Weaknesses:

This paper argues that the proposed new strategy can be used to prove the convergence of deep learning architectures to a zero training".  However, only two-layer NNs are discussed in case studies. Can you provide some cases to support your argument on ``the deep architectures"?

---

> ### Author Response · Authors · 2022-08-02
> **Response to Reviewer csDk**
>
>
> Thank you for your time and thoughtful comments.
> We have updated our submission to address your concerns (pdf in updated supplementary material, changes in red).
>
> Regarding minor concerns, we have added a notation table in Appendix A.1, and hope this helps with the readability, please let us know if you find that other shorthands deserve to be added to this table or need more attention. We took $\mu$-uniform-conditioning of ${K_\theta^\star}$ to mean that the lowest eigenvalue is at least $\mu$, i.e. ${K_\theta^\star} \succeq \mu$ (as positive semi-definite matrices). We have removed this term from the introduction to avoid misunderstandings, and replaced it with a more explicit description.
>
> The question of the gap from presented results to deep networks is harder to address.
> As a first and high-level answer, we use the term "deep learning" in this sentence in a broader sense, to mean essentially gradient-based iterative training of machine learning models in the massive-data regime, which is a looser definition but arguably compatible with a more modern use of the term, although not directly connected to the layer-depth of the model which initially led to this choice of terminology. In this broad sense, we believe that any theory powerful enough to explain the vanishing optimization error of deep learning will have to recover nearly-trivial systems such as the lemniscate as subcases. This example shares some properties with layer-deep architectures that rule out both parameter-convex and definite-NTK theories. We argue not that this approach will surely lead to convergence proofs of deep architectures, but only that previous approaches won't and show why. Our approach is then in a way incomplete by construction, but can be viewed as the easy case that later theories must recover as a subcase to stand a chance, which substantially shrinks the search space for such theories.
>
> Regarding a stricter interpretation of the term "deep" learning, and the road to transformers and the likes,
> we still have high hopes for this line of work.
> We have started exploring some ideas to extend the result from two-layer networks to deeper multi-layer perceptrons, but don't have fully fleshed-out results at the moment that could be added to this submission. Should this submission be accepted, we will consider extending it with more examples in the direction you suggest in a follow-up journal version, to strengthen this argument.

---

> > ### Comment · Reviewer_csDk · 2022-08-08
> > **A very insightful work.**
> >
> > I agree with that there is a gap from the presented results to deep networks but it is still a very insightful work. I am looking forward to your follow-up version.

---

### Meta-Review · Area_Chair_kidX · 2022-08-29

**Recommendation:** Accept
**Confidence:** Less certain

**Metareview:**

This paper proposes a new method for proving the convergence of gradient flow to zero loss by leveraging Rayleigh quotients to establish KL inequalities, a strategy that can apply even without overparameterization. The reviewers found the paper to be well written and generally easy to follow, despite a few concerns about burdensome notation. The discussion highlighted a few minor technical issues which were addressed in the revision. Overall the consensus is that this paper provides valuable new tools and the results will be of interest to the theory community, so I recommend acceptance.


**Award:**

No

---

### Decision · Program_Chairs · 2022-09-14

Accept